# Metabolic dependencies of metastasis-initiating cells in female breast cancer

C. Megan Young[1,2,3], Laurent Beziaud[1,2,3], Pierre Dessen[1,2,3], Angela Madurga Alonso [1,2,3], Albert Santamaria-Martínez [1,3] ✉ & Joerg Huelsken [1,2,3] ✉

Understanding the mechanisms that enable cancer cells to metastasize is essential in preventing cancer progression. Here we examine the metabolic adaptations of metastasis-initiating cells (MICs) in female breast cancer and how those shape their metastatic phenotype. We find that endogenous MICs depend on the oxidative tricarboxylic acid cycle and fatty acid usage. Sorting tumor cells based upon solely mitochondrial membrane potential or lipid storage is sufficient at identifying MICs. We further identify that mitochondrially-generated citrate is exported to the cytoplasm to yield acetyl-CoA, and this is crucial to maintaining heightened levels of H3K27ac in MICs. Blocking acetyl-CoA generating pathways or H3K27ac-specific epigenetic writers and readers reduces expression of epithelial-to-mesenchymal related genes, MIC frequency, and metastatic potential. Exogenous supplementation of a short chain carboxylic acid, acetate, increases MIC frequency and metastasis. In patient cohorts, we observe that higher expression of oxidative phosphorylation related genes is associated with reduced distant relapse-free survival. These data demonstrate that MICs specifically and precisely alter their metabolism to efficiently colonize distant organs.

Cancer cells exhibit altered biosynthetic and energy requirements which result in heightened and biased fuel uptake and utilization. Blocking required metabolic alterations such as increased glucose and/or amino acid utilization can effectively blunt transformation by inhibiting proliferation and causing cell death[1–3]. Metabolism not only has supportive and enabling functions, but it can also directly influence cellular phenotypes. For example, hypoxia can induce cancer invasion[4] and the availability of certain amino acids can alter drug response[5,6]. This demonstrates that changes in metabolism are essential in both supporting and driving cancer progression.

Metastatic disease causes the majority of cancer deaths and there are still very few therapeutic options for patients that are diagnosed at this late stage. Thus, understanding mechanisms for metastatic dissemination and progression can uncover actionable vulnerabilities and potentially improve therapy. In many tumor types, metastasis is caused by a rare population of stem-like cells in the heterogenous tumor mass that are highly plastic. During metastasis formation, these metastasis-initiating cells (MICs) are required to undergo sequential, partially opposing phenotypic changes that enable them to invade into the surrounding tissue, enter and survive in the bloodstream, and generate a new tumor at a distant site[7]. MICs have been identified in a variety of solid tumors, such as breast cancer[8,9], pancreas cancer[10], colorectal cancer[11], and oral squamous cell carcinoma[12], and their frequency has been shown to be correlated with poor prognosis.

In addition, comparisons of genome sequences between MICs and their non-MIC counterparts show a high degree of concordance[13,14], suggesting that the ability to metastasize to distant organs is not likely due to the accumulation of metastasis-specific driver mutations, but rather due to epigenetic modifications that enable reversible transitions in cell phenotype during the course of metastasis. In line with

[1]École Polytechnique Fédérale de Lausanne (EPFL), ISREC (Swiss Institute for Experimental Cancer Research), 1015 Lausanne, Switzerland. [2]Agora Cancer Research Center, Rue du Bugnon 25A, 1011 Lausanne, Switzerland. [3]Swiss Cancer Center Léman, Lausanne, Switzerland. ✉e-mail: albert.santamariamartinez@epfl.ch; joerg.huelsken@epfl.ch

this, alterations in certain histone marks have recently been shown to be necessary and sufficient for cell state changes such as the epithelial-to-mesenchymal transition (EMT)[15,16]. Some non-MICs are even able to dedifferentiate into MICs by maintaining key genes in a poised chromatin state[17], demonstrating that epigenetic modifications endow cells with the plasticity to adapt to various requirements during the metastatic cascade[18].

Metabolic alterations have been linked to each step of metastasis. For example, reductive glutamine metabolism is required to generate antioxidants for combatting reactive oxygen species under low attachment conditions, such as when traveling through the bloodstream[19], and pyruvate is essential for generating sufficient levels of alpha-ketoglutarate for collagen stabilization at the metastatic site for efficient tissue colonization[20]. This raises the possibility that the metabolic phenotype of MICs is constituent for its metastatic potential.

Metabolism and epigenetics are inextricably linked. In contrast to kinases whose activity is controlled by upstream signaling pathways and not the abundance of its substrate, ATP, the activity of various epigenetic enzymes is influenced by the concentrations of certain metabolites which can be cofactors, substrates, or allosteric regulators for these enzymes[21]. Reduced availability of s-adenosyl methionine for example, has been shown to reduce histone methylation and increase subsequent transcription of genes in oncogenic signaling pathways[22,23].

Here we aim to investigate the metabolic differences between MICs and non-MICs and whether that may influence the transcription of gene programs in MICs that enable them to generate metastatic lesions. To do this, we utilize tumors deriving from the autochthonous breast cancer model, MMTV-Polyoma Middle T (PyMT), since this is where we had previously identified endogenous MICs that are exclusively capable of generating metastasis[8]. We also validate and extend these findings in a second mouse model using the highly metastatic triple negative breast cancer (TNBC) cell line 4T1. We uncover that MICs rely on fatty acid oxidation (FAO) and the oxidative tricarboxylic acid (TCA) cycle, and that these contribute to acetyl-CoA generation and the acetylation of histones on EMT-related genes. Blocking either the metabolic pathways that generate acetyl-CoA or acetylation reactions can reduce MIC frequency and metastatic potential in vivo.

## Results

### Metastasis-initiating cells show increased mitochondrial activity
Considering the conflicting evidence in the literature for the role of mitochondrial activity during metastasis[24,25], we sorted 4T1 breast cancer cells for varying levels of mitochondrial activity based on a fluorescent stain that preferentially accumulates in mitochondria with high mitochondrial membrane potential (MMP). High MMP cells have increased intracellular ATP compared to their low MMP counterparts[26] (Supplementary Fig. 1a). Tail vein injections (TVIs) revealed that the cells with high MMP generate 10x and 5x more metastatic nodules in the lungs compared to their low and medium counterparts, respectively (Fig. 1a), despite having similar proliferation rates (Supplementary Fig. 1b). We observed a similar MMP phenotype in 4T1 MICs that are ALDH^bright compared to ALDH^dim non-MICs[27] (Fig. 1b and Supplementary Fig. 1c). To confirm this phenotype in another model, we utilized the PyMT breast cancer model where we had previously identified metastasis-initiating cells (MICs) as Lin⁻CD24⁺CD90⁺[8]. Using the same mitochondrial stain, we observed that PyMT MICs have an almost 2-fold increase of MMP compared to their non-MIC counterparts (Lin⁻CD24⁺CD90⁻; Fig. 1c and Supplementary Fig. 1d), even when normalizing for mitochondrial mass (Supplementary Fig. 1e). RNA-sequencing of sorted PyMT MICs and non-MICs revealed increased expression of gene sets related to oxidative phosphorylation (OXPHOS) and mitochondrial metabolism (Fig. 1d), in addition to the expected increased expression of gene sets related to EMT and partial EMT[28] (Supplementary Fig. 1f). These data suggest that in metastatic

cells, fuels may be preferentially utilized in mitochondria. To test this hypothesis, we measured oxygen consumption rate (OCR) and extra-cellular acidification rate (ECAR) in a Seahorse mitochondrial stress test. It was not possible to use sorted populations for this since we observed poor metabolic health and up to three-fold reduction of OCR in sorted cells (Supplementary Fig. 1g). Instead, we used tumorsphere- and monolayer-grown cells, since tumorspheres are known to be enriched for highly tumorigenic stem-like cells[29]. When only glucose was provided as a fuel source, we observed that tumorspheres have a higher maximal respiratory capacity (Fig. 1e) and conversely lower glycolytic capacity (Fig. 1f), confirming our hypothesis. We also observed that tumorsphere-derived cells have a higher MMP compared to their monolayer counterparts (Supplementary Fig. 1h). Driving monolayer grown cells to undergo OXPHOS by replacing glucose in the cell culture media with galactose, which does not yield ATP through glycolysis and thereby shifts to the use of OXPHOS, increased MIC frequency in PyMT cells (Fig. 1g) and metastatic capacity in 4T1 cells (Fig. 1h). The increase of MICs we observed is not due to preferential death of non-MICs under nutrient stress as measured by Annexin V and propidium iodide staining (Supplementary Fig. 1i), or preferential increase in proliferative capacity of MICs as measured by reduction of CellTrace Violet staining (Supplementary Fig. 1j), but rather due to the maintenance of the MIC phenotype and/or the conversion of non-MICs to MICs (Supplementary Fig. 1k). Collectively, these data show that metastatic cells exhibit higher mitochondrial activity which directly affects their ability to colonize distant organs.

### Metastasis initiating cells are dependent on enhanced fatty acid flux and usage
We then investigated if MICs have a preferential fuel source by performing a Seahorse-based fuel dependency test, which compares the change in OCR when first blocking the entry of one of the three main fuel sources into the mitochondria (pyruvate, glutamine, or long chain fatty acids (LCFAs)), then subsequently blocking the remaining two. We observed that in comparison to monolayer grown cells, tumor-spheres do not differ in their dependency on either pyruvate or glutamine (Supplementary Fig. 2a), however, tumorspheres have an almost 2-fold increased dependency for LCFA oxidation (Fig. 2a). We then confirmed that this dependency is not due to a reduced capacity for LCFA oxidation in the monolayer cells because if we force the cells to oxidize LCFAs by first simultaneously blocking entry of pyruvate and glutamine into the mitochondria, we do not observe differences between the monolayer cells in comparison to the tumorspheres (Supplementary Fig. 2b).

We confirmed the dependency on LCFA oxidation by performing experimental metastasis assays via TVI with PyMT cells that had been pre-treated for 72 h with etomoxir, an inhibitor of the rate limiting enzyme in FAO, CPT1. Despite not significantly altering proliferative potential (Supplementary Fig. 2c), etomoxir reduced both MIC frequency (Fig. 2b) and metastatic capacity (Fig. 2c), and these results were confirmed using miR-mediated knockdowns (KDs) of Cpt1a and Cpt2 (Fig. 2d and Supplementary Fig. 2d). This was further validated by performing spontaneous metastasis assays where orthotopically implanted tumors that express miR-mediated KDs of either Gfp or Cpt1a were resected just before reaching 1000 mm³. Since the tumors were growing at different rates, we chose to perform tumorectomies at a fixed size, rather than a fixed time to give slower growing tumors more time to metastasize (Supplementary Fig. 2e). We then analyzed the lungs 30 days after their respective tumor removal to examine the number of spontaneous metastatic nodules formed. We again observed a significant reduction of lung metastasis upon Cpt1a KD (Supplementary Fig. 2f). In order to confirm that this dependency was specific to lipids, TVI of cells that were pre-treated with UK5099, an inhibitor of the mitochondrial pyruvate carrier, did not change metastatic capacity (Supplementary Fig. 2g).

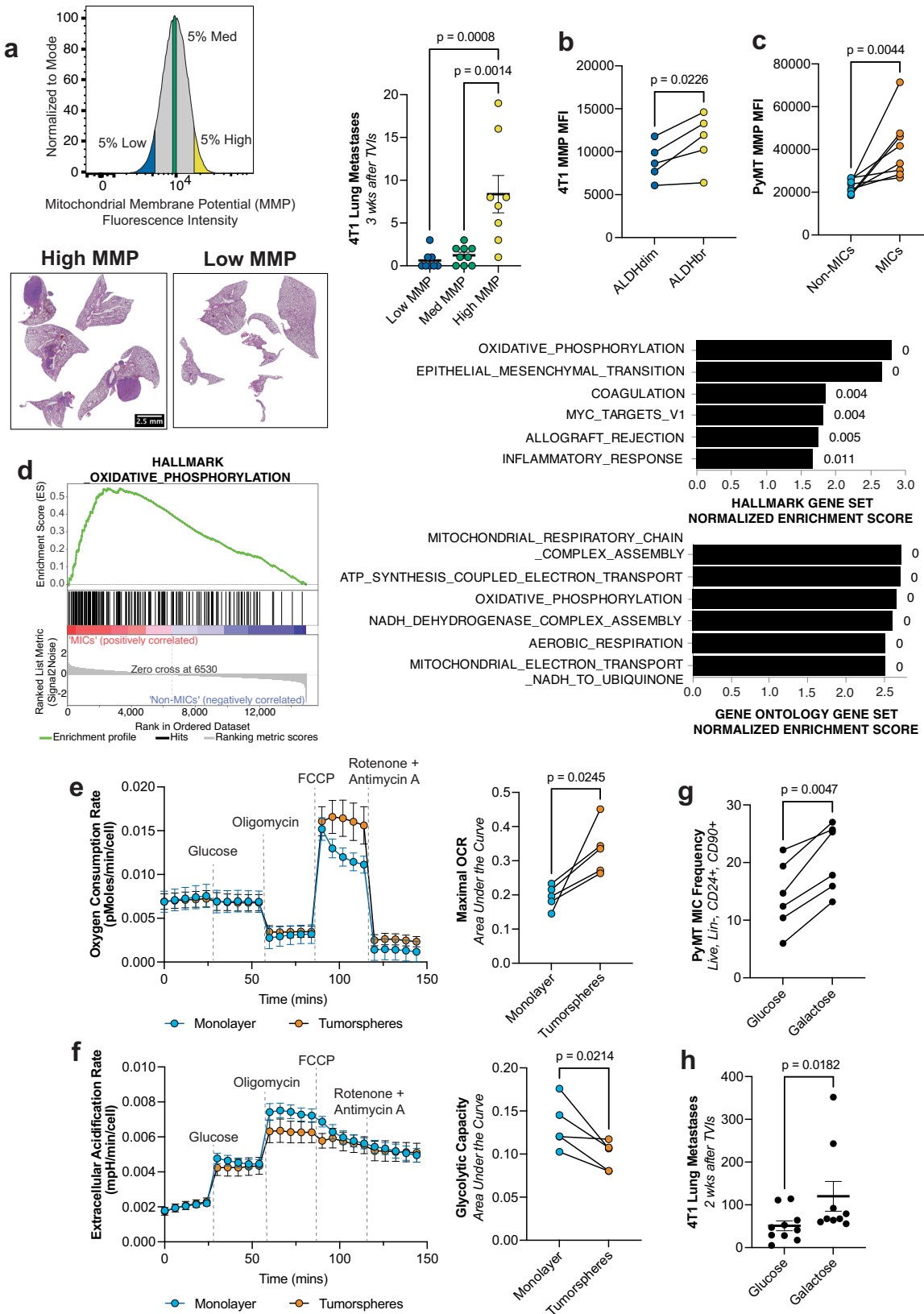

Using fluorescently tagged lipid probes ex vivo, we observed that MICs from both orthotopically implanted 4T1 tumors and spontaneous PyMT tumors have a 2-fold and over 30% increase in LCFA uptake (Fig. 2e, f) and a 3-fold and 30% increase of lipid stores, respectively (Fig. 2g, h). By sorting 4T1 cells that have high, medium, and low levels of lipid stores, we confirmed that cells that have the highest levels of lipid stores also have the highest propensity to generate metastatic nodules (Fig. 2i), despite similar proliferation rates (Supplementary Fig. 3a). Likewise, in the early stage of metastasis formation (3 days after TVI), there was an over 4-fold enrichment of cells with elevated lipid stores observed by oil red O staining, compared to subsequent stages of metastasis formation (5 days after TVI

**Fig. 1 | Heightened mitochondrial activity is a key feature of MICs. a** Lung metastases in recipient wildtype (wt) Balb/c after TVI of 4T1 cells sorted for low, med, and high MMP. Left top: gating strategy for sorting cells. Left bottom: representative images of hematoxylin and eosin paraffin sections of lungs of recipient mice. Right: quantification. ($n = 8$, ordinary one-way ANOVA followed by Tukey's test). **b** MMP MFI in ALDH$^{dim}$ vs. ALDH$^{bright}$ 4T1 cells. ($n = 5^a$, paired $t$-test). **c** MMP MFI in ex vivo PyMT non-MICs vs. MICs. ($n = 8^a$, paired $t$-test). **d** Upregulated gene sets when comparing ex vivo PyMT MICs vs. non-MICs. Left: gene set enrichment plot for the HALLMARK_OXIDATIVE_PHOSPHORYLATION signature from the Molecular Signature Database (MsigDB). Right: top upregulated gene sets from the hallmark (top) and gene ontology (bottom) collections. Numbers beside bar plots indicate false discovery rate (FDR). ($n = 3^a$). **e** OCR over time in tumorsphere- vs. monolayer-grown PyMT cells (normalized to number of cells). Right:

OCR area under the curve (AUC) for matched monolayer and tumorsphere-grown PyMT cells. ($n = 5^a$, paired $t$-test). **f** Extracellular acidification rate over time during the mito stress test in tumorsphere- vs. monolayer-grown PyMT cells (normalized to number of cells). Right: glycolytic capacity AUC for matched monolayer- and tumorsphere-grown PyMT cells. ($n \geq 3^a$, paired $t$-test). **g** PyMT MIC frequency after culture in either glucose or galactose containing media. ($n = 6^a$, paired $t$-test). **h** Number of lung metastases in recipient wt Balb/c mice after TVI with 4T1 cells cultured in either glucose- or galactose-containing media. ($n = 10$ for glucose, $n = 9$ for galactose, Mann–Whitney test). MFI mean fluorescence intensity. Paired $t$-tests were two-tailed and by ratio. Values shown correspond to means ± SEM. Source data are provided in the source data file. $^a$ signifies number of independent experiments or tumors.

and onwards; Fig. 2j and Supplementary Fig. 3b). In PyMT cells under starved conditions, lipid droplets are used up more rapidly in MICs than non-MICs (Supplementary Fig. 3c). This becomes more evident if lipid droplet generation is further blocked using T863 & PF-06424439 to inhibit the lipogenic enzymes DGAT1 & DGAT2, respectively, in starved conditions compared to complete media conditions (Supplementary Fig. 3d). TVIs indicate that blocking the generation of these lipid stores even under nutrient replete conditions prevents metastasis (Fig. 2k) without compromising proliferative potential (Supplementary Fig. 3f). On the other hand, if breakdown of lipid stores is inhibited using the ATGL inhibitor, ATGListatin, in complete media conditions, there is an accumulation of lipid stores (Supplementary Fig. 3e), but metastatic activity is inhibited (Fig. 2l) without altering proliferation (Supplementary Fig. 3f). In summary, our results reveal enhanced and dynamic flux of lipids (uptake, storage, and oxidation) in MICs compared to non-MICs which is crucial for metastasis generation.

## Modulating acetyl-CoA generation and availability alters metastatic capacity

The data thus far showed that MICs preferentially utilize OXPHOS in contrast to glycolysis with enhanced dependency on FAO. Davis and collaborators had suggested that ATP generation may limit metastasis[30]. However, when PyMT and 4T1 cells were pre-treated with oligomycin to block mitochondrial ATP generation, metastasis formation was not reduced (Supplementary Fig. 4a, b). Further, when comparing oxygen consumption rates (OCR) as a proxy for ATP generation between tumorspheres and monolayer grown cells, we also observed no differences (Supplementary Fig. 4c). When inhibiting aconitase 2 (ACO2) which converts citrate to isocitrate in the mitochondrial TCA cycle (Fig. 3a) using fluoroacetate as pre-treatment, this increased both MIC frequency and metastatic capacity in both PyMT (Fig. 3b, c) and 4T1 cells (Supplementary Fig. 4d). Fluoroacetate causes accumulation of citrate[31] (Supplementary Fig. 4e) which can be exported from mitochondria and converted to acetyl-CoA by ACLY. Using an inhibitor against ACLY (SB204990) as pre-treatment, this resulted in a significant reduction of MIC frequency (Fig. 3d) and metastasis in the PyMT model (Fig. 3e), and reduced metastasis when combined with etomoxir in the 4T1 model (Fig. 3f). These results were confirmed using miR-mediated KDs in both models (Fig. 3g, h). This reduction in metastasis and MIC frequency by ACLY inhibition was rescued by the combined treatment with fluoroacetate (Fig. 3i, j and Supplementary Fig. 4f). Due to the reduction of metastasis we observed by blocking ACLY, we then hypothesized that the generation of acetyl-CoA may be limiting for metastatic colonization. To test this, we increased the availability of acetyl-CoA by adding acetate, which gets converted into acetyl-CoA by ACSS1 and ACSS2, to the cell culture media (Fig. 3a). Acetylcarnitine, which has also been shown to support histone acetylation by serving as another acetyl-CoA precursor[32], was confirmed to increase upon acetate treatment (Supplementary Fig. 4g). Cells that had been pre-treated with acetate prior to TVI,

showed an increase of MIC frequency and metastasis, both in the PyMT (Fig. 3k, l) and the 4T1 models (Supplementary Fig. 4h). Further, TVI of cells with KDs of *Acss1* and *Acss2* resulted in reduced metastasis formation (Supplementary Fig. 4i, j). In all the cases described, we did not observe significant alterations in proliferation upon treatment or KD (Supplementary Fig. 4k–n). Overall, these data indicated that enhanced mitochondrial activity in MICs may reflect an increased requirement for acetyl-CoA derived from citrate released from mitochondria into the cytoplasm.

## MICs show enhanced requirement for acetyl-CoA supply to control histone acetylation

Since acetyl-CoA is a cofactor for histone acetylation, we sought to examine if the reliance of MICs on acetyl-CoA may be due to global alterations in epigenetic modifications between MICs and non-MICs. We observed a significant increase of the activating mark of promoters and enhancers, H3K27ac, in endogenous PyMT MICs vs. non-MICs (Fig. 4a). In the 4T1 model, we also observed that the cells that have the highest MMP and metastatic potential are also those that have the highest H3K27ac levels (Fig. 4b). Further, if we pre-treat cells with inhibitors that block writers and readers of H3K27ac (C646 for p300 and JQ1 for BET family members, respectively) and again perform TVI, we can significantly reduce the MIC frequency and number of metastases formed (Fig. 4c–e) without significantly impacting proliferative potential (Supplementary Fig. 5a).

We therefore extended these measurements of H3K27ac to treatments that we had identified to reduce metastatic capacity and are involved in acetyl-CoA generation. In the PyMT model, we observed a significantly reduced level of H3K27ac upon *Acly* and *Cpt1a* inhibition either by pharmacological inhibitors (Fig. 4f) or miR-mediated KDs (Supplementary Fig. 5b). In contrast, acetate treatment, which increases metastasis, also increased H3K27ac (Fig. 4f). Similarly, in the 4T1 model, H3K27ac increased upon acetate treatment, reduced upon combined etomoxir and SB204990 treatment, and the combination of all three rescued the acetylation levels back to vehicle-treated controls (Fig. 4g). In addition, we can prevent the acetate-induced increase of MIC frequency in the PyMT model (Fig. 4h, i, k) and acetate-induced metastatic capacity in the 4T1 model with the addition of etomoxir, SB204990, or JQ1 (Fig. 4j, l and Supplementary Fig. 5c, d) without severely impacting proliferative potential (Supplementary Fig. 5e). These data suggest that in MICs, the supply of acetyl-CoA may be rate-limiting in controlling epigenetic events which drive the metastatic cascade. Since there are many sources of acetyl-CoA in cells (lipid catabolism, citrate export from mitochondria to the cytoplasm, uptake of short chain fatty acids, recycling of acetyl groups after histone deacetylation, etc.) with often limited metabolic capacity, the contributions deriving from individual sources are likely additive and may partially compensate for each other. As we show now, increased mitochondrial activity aids metastasis, at least in part, by adding to the acetyl-CoA pool to allow for histone acetylation.

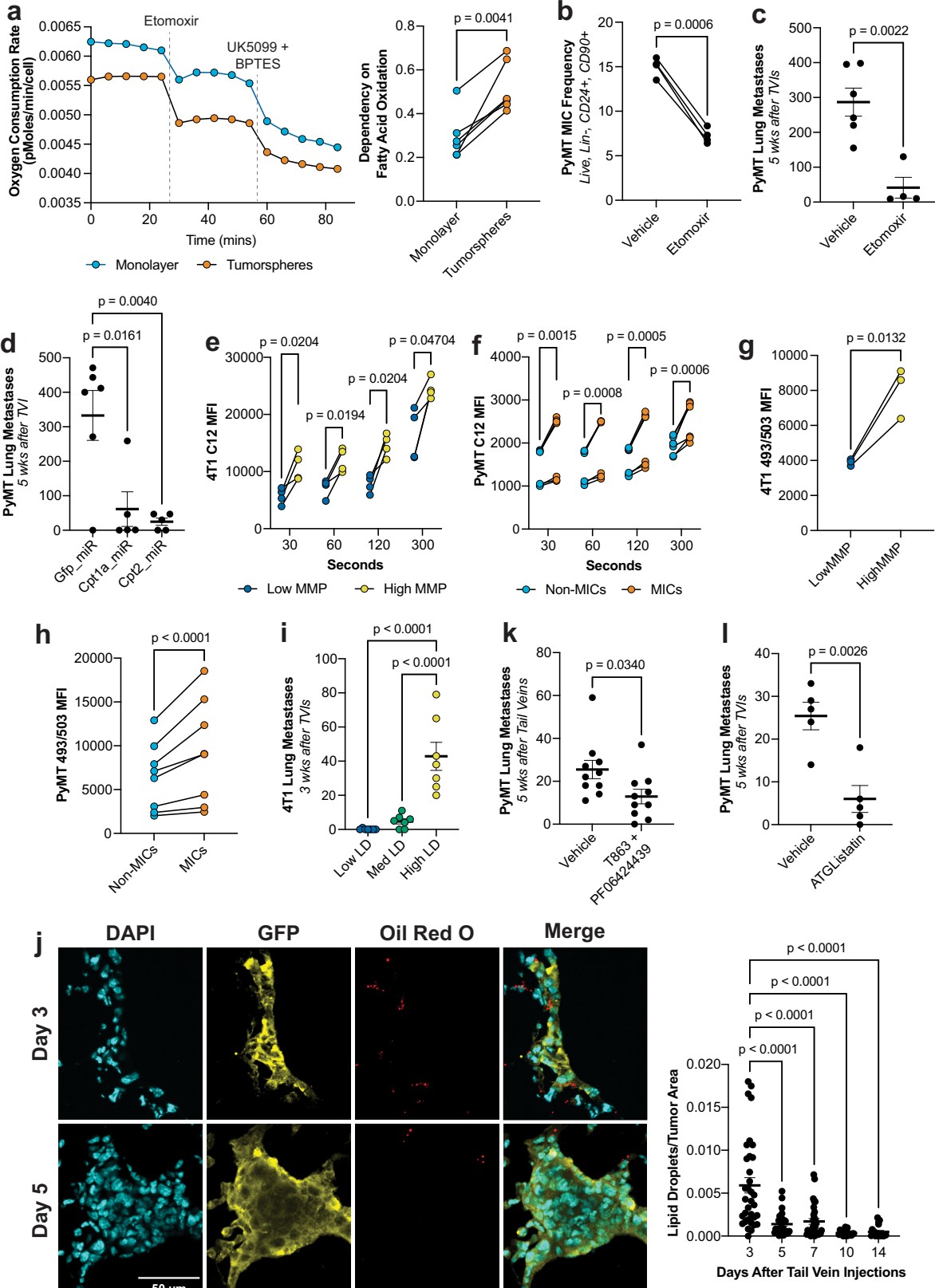

## Increased acetylation on H3K27 induces expression of genes for epithelial-to-mesenchymal transition

To determine how increased acetyl-CoA generation boosts metastasis formation, we performed RNA-sequencing on acetate- vs. vehicle-treated 4T1 cells. Gene-set enrichment analysis revealed that in acetate-treated cells the highest scoring Hallmark gene set is EMT (Fig. 5a), and several of the highest scoring Gene Ontology gene sets are related to extracellular matrix remodeling (Supplementary Fig. 6a). We also observed the upregulation of the Hallmark EMT gene set in high MMP cells compared to low MMP cells (Supplementary Fig. 6b). We validated the acetate-induced phenotype by performing qPCR of known EMT markers such as collagens, matrix metalloproteases, and

**Fig. 2 | MICs rely on fatty acid turnover and usage. a** Left: normalized OCR over time in tumorsphere- vs. monolayer-grown cells. Right: dependence on LCFA oxidation calculated as described in the methods ($n \geq 3^a$, paired $t$-test). **b** PyMT MIC frequency after treatment with etomoxir. ($n = 4^a$, paired $t$-test). **c** Number of lung metastases in recipient wildtype (wt) FVB/N mice after TVI of PyMT cells treated with etomoxir. ($n = 5$, unpaired $t$-test). **d** Number of lung metastases in recipient mice after TVI of PyMT cells containing miR-mediated KDs of *Gfp*, *Cpt1a*, and *Cpt2*. 84% KD efficiency for *Cpt1a* and *Cpt2*. ($n \geq 5$, unpaired $t$-test). **e** LCFA uptake measured by MFI of 4T1 cells sorted for low and high MMP after BODIPY FL C12 staining for specified durations. ($n = 4^a$, paired $t$-test). **f** LCFA uptake measured by MFI of ex vivo PyMT non-MICs vs. MICs after BODIPY FL C12 staining for specified durations. ($n = 7^a$, paired $t$-test). **g** Lipid stores measured by MFI of 4T1 cells sorted for low and high MMP after BODIPY 493/503 staining. ($n = 3^a$, paired $t$-test). **h** Lipid stores measured by MFI of ex vivo PyMT non-MICs vs. MICs after BODIPY 493/ 503 staining. ($n = 3^a$, paired $t$-test). **i** Number of lung metastases in recipient wt Balb/c mice after TVI of 4T1 cells sorted for low, med, and high levels of lipid droplets. ($n = 7$, ordinary one-way ANOVA followed by Tukey's test). **j** Number of lipid droplets measured by oil red O, normalized to tumor area for the specified timepoints after TVI of GFP + 4T1 cells into recipient DEREG mice. Left: representative images from each timepoint. Right: quantification ($n \geq 18$, one-way ANOVA followed by Dunnett's test). **k** Number of lung macrometastases in recipient mice after TVI of PyMT cells treated with T863 and PF-06424439. ($n = 10$, unpaired $t$-test). **l** Number of lung macrometastases in recipient mice after TVI of PyMT cells treated with ATGListatin ($n = 5$, unpaired $t$-test). MFI mean fluorescence intensity. Paired $t$-test were two-tailed and by ratio. Unpaired $t$-test were parametric and two-tailed. Values shown correspond to means ± SEM. Source data are provided in the source data file. $^a$ signifies number of independent experiments or tumors.

integrins. Not only did we see the expected increase of EMT markers in the acetate treatment, but when in combination with either etomoxir, SB204990, etomoxir + SB204990, or JQ1, the expression of these EMT markers returned to baseline levels (Fig. 5b and Supplementary Fig. 6c).

To confirm that these gene expression changes are due to increased H3K27ac acetylation, we performed an H3K27ac ChIP-Seq on 4T1 cells that had been treated with acetate, etomoxir, etomoxir + acetate, etomoxir + SB204990, and etomoxir + SB204990 + acetate. We observed increases in the acetylation peaks of several EMT genes of interest such as *Postn, Col1a1, Sparc, and Itga5* in the acetate-treated cells compared to vehicle-treated cells, and reductions in the same genes upon treatment with etomoxir or etomoxir + SB204990 (Fig. 5c and Supplementary Fig. 6d). Additionally, the reduced peaks that were identified to be associated with EMT genes increased upon combination with acetate (Fig. 5d and Supplementary Data 1).

We therefore propose a model wherein MICs upregulate OXPHOS and FAO that contribute to citrate and acetyl-CoA generation for heightened H3K27ac. This then enables MICs to increase transcription of EMT-related genes and facilitates metastasis (Fig. 5e).

### Human breast cancer cell lines exhibit a similar dependence on mitochondrial activity and acetyl-CoA for metastasis

To determine if we can observe similar phenotypes in human breast cancer, we first sorted human TNBC cells, MDA-MB-231 and BT549, for those with high and low MMP. Just as we observed in the mouse models, the cells with high MMP also had higher levels of lipid uptake (Fig. 6a, b) and storage (Fig. 6c) in comparison to those with low MMP.

We were also able to reduce the ability of human TNBC cell lines to metastasize by knocking down enzymes that generate acetyl-CoA. We utilized miR-mediated KDs against *ACLY* and *ACSS2* in MDA-MB-231 and observed a significant reduction of metastasis (Fig. 6d) without severely compromising proliferative potential (Supplementary Fig. 7a). In HCC1806 cells, however, we observed no change in metastatic capacity upon KD of ACLY (Supplementary Fig. 7b). Considering we had observed a two-fold increase of *ACSS2* expression when knocking down *ACLY* in both HCC1806 and BT20 cells (Supplementary Fig. 7c), we surmised that *ACSS2* was acting to compensate for the loss of *ACLY*. *ACSS2* converts acetate to acetyl-CoA in the cytoplasm and nucleus and has been shown to be upregulated when *ACLY* is deficient to maintain intracellular levels of acetyl-CoA[33]. We therefore performed TVI with BT20 and HCC1806 cells containing miR-mediated KDs of both *ACLY* and *ACSS2* to avoid this compensation. In this case, we observed that metastasis numbers were reduced (Fig. 6e and Supplementary Fig. 7d), again, without affecting proliferation (Supplementary Fig. 7e, f). Finally, upon TVI of acetate-treated MDA-MB-231 and BT20 cells, we observed significantly increased numbers of metastatic nodules in the lungs (Fig. 6f and Supplementary Fig. 7g).

We then focused on analyzing previously published datasets by the Cancer Cell Line Encyclopedia (CCLE), which had characterized 42 combinations of histone marks in about 900 cell lines[34] and measured the metastatic potential of over 500 cancer cell lines (MetMap)[35]. We focused on the MetMap breast cancer cohort where 21 barcoded TNBC cell lines were examined for their metastatic potential. Upon comparing these epigenetic patterns to their respective metastatic potential, we observed that only histone marks containing H3K27ac were repeatedly significantly correlated with metastatic potential (Fig. 6g, h and Supplementary Table 1).

Next, we looked at breast cancer patient cohorts to determine if the OXPHOS signature identified in our PyMT MICs vs. non-MICs is predictive of distant relapse-free survival. We therefore stratified a patient cohort comprising of 50 individual microarray datasets based on their expression of just under 100 OXPHOS signature genes (Supplementary Data 2). Consistent with our hypothesis, patients that had higher expression of this signature were more likely to develop distant relapse (Fig. 6i). This is also evident when performing the analysis only in patients with basal breast cancer as classified by PAM50 (Supplementary Fig. 7h). For patients who have not yet received systemic treatment, those who express higher levels of ACLY mRNA are also more likely to develop distant relapse (Supplementary Fig. 7i). Moreover, in both the microarray cohort and the TCGA cohort, patients who have higher expression of CPT1A have a reduced probability of overall survival (Supplementary Fig. 7j, k). These data reveal a correlation between mitochondrial activity, lipid usage, acetyl-CoA generation, and disease progression in breast cancer that is not only evident in our experimental mouse settings, but also in patients.

## Discussion

Dissecting the mechanisms that endow MICs with the ability to metastasize is essential to preventing cancer progression and potentially improving available therapies. In this study, we identified a metabolic program unique to MICs which contributes to their metastatic phenotype. We found that endogenous MICs in breast cancer depend on high mitochondrial activity in part fueled by FAO and export of citrate from mitochondria to generate acetyl-CoA. Acetyl-CoA is required for histone acetylation which is globally elevated in MICs and drives expression of EMT-related genes. Blocking either metabolic pathways or epigenetic modifiers that interrupt this process reduces the metastatic capacity of MICs, while supplementing with an exogenous source of acetyl-CoA enhances metastatic capacity.

High mitochondrial activity is constituent to the MIC phenotype. We show for the first time that by sorting cells for just one parameter, high MMP, we can identify the cells that are solely responsible for generating metastatic lesions. Further by forcing cells to undergo OXPHOS, we can convert non-MICs to MICs and increase metastasis. Our results extend previous work that demonstrates that cells devoid of mitochondria need to reacquire them from their host tissue for tumor initiation and metastasis[36,37]. It has been postulated by other groups that mitochondrial metabolism is important for metastasis due to the heightened ATP requirement[30,38], but we show here that pre-

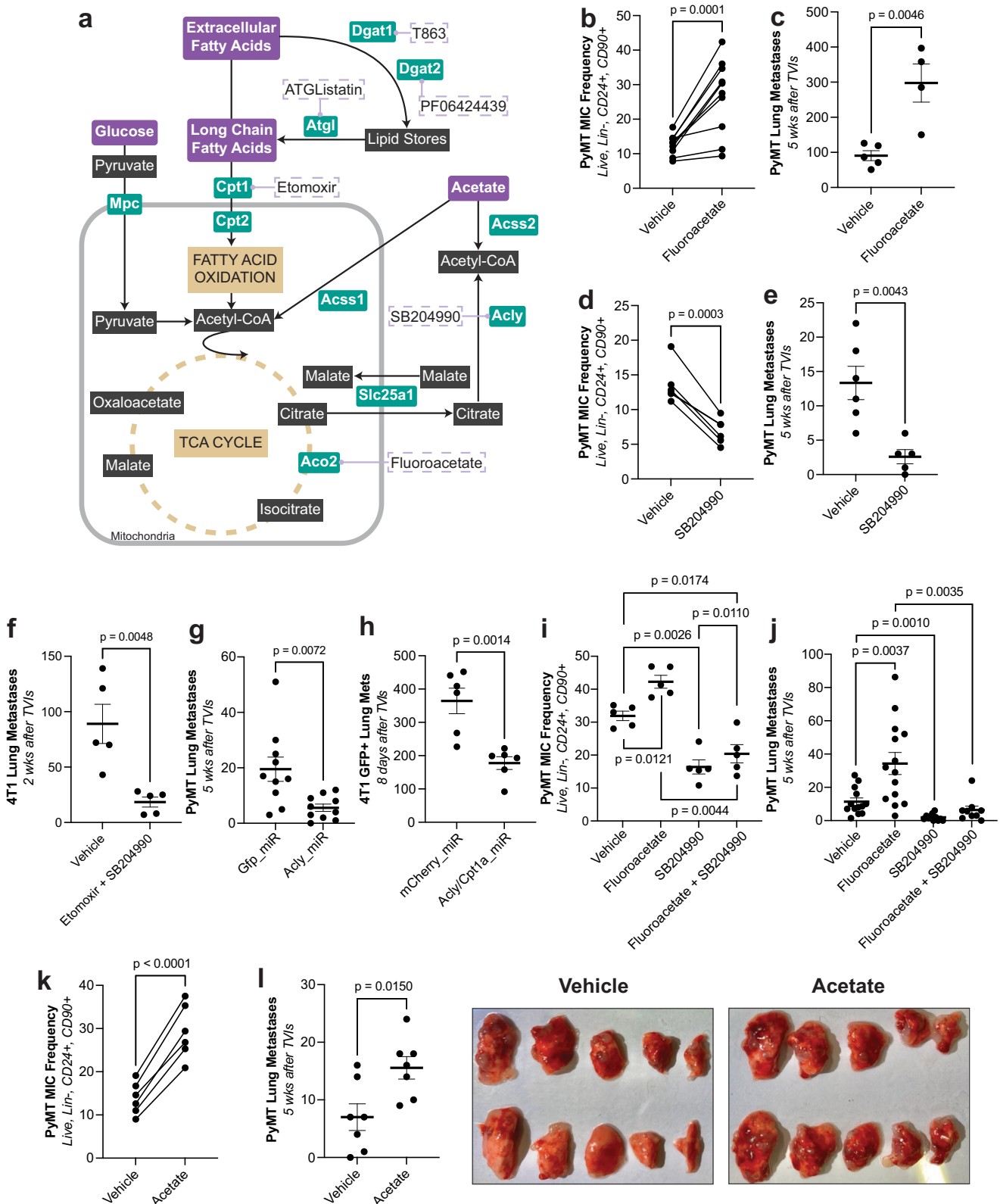

treatment of cells with oligomycin, which blocks mitochondrial ATP generation, does not reduce the ability of MICs to metastasize. Instead, we observe that the high mitochondrial activity is correlated with increased H3K27ac and increased expression of EMT-related genes.

We also show that this high mitochondrial activity is fueled at least in part by heightened FAO in MICs. Considering that we also observed high levels of H3K27ac in MICs, we hypothesized that the increased FAO

could contribute to heightened histone acetylation in MICs. Glucose has been canonically shown to be the primary nutrient that contributes to histone acetylation[39], however, it has also been shown that medium-chain fatty acids are able to contribute up to 90% of the carbon for histone acetylation in a mouse hepatocyte cell line[40]. We now show that blocking the entry of LCFAs into the mitochondria by genetic or pharmacological means can not only reduce histone acetylation on H3K27,

**Fig. 3 | MICs rely on pathways that generate acetyl-CoA. a** Diagram of the relevant metabolic pathways and their respective inhibitors. **b** PyMT MIC frequency after treatment with fluoroacetate. ($n = 10$[a], paired $t$-test). **c** Lung metastases in recipient wildtype (wt) FVB/N mice after TVI of PyMT cells treated with vehicle or fluoroacetate. ($n = 5$, $n = 4$, respectively, unpaired $t$-test). **d** PyMT MIC frequency after treatment with vehicle or SB204990. ($n = 6$[a], paired $t$-test). **e** Lung metastases in recipient wt FVB/N mice after TVI of PyMT cells treated with vehicle or SB204990. ($n = 6$, $n = 5$, respectively, unpaired $t$-test). **f** Lung metastases in recipient wt Balb/c mice after TVI of 4T1 cells treated with vehicle or etomoxir and SB204990. ($n = 5$, unpaired $t$-test). **g** Lung metastases in recipient wt FVB/N mice after TVI of PyMT cells containing miR-mediated KDs of *Gfp* or *Acly*. 77% KD efficiency for *Acly*. ($n = 10$, unpaired $t$-test). **h** GFP+ lung metastases in recipient

Rag2$^{-/-}$γc$^{-/-}$ mice after TVI of GFP + 4T1 cells containing miR-mediated KDs of *mCherry* or *Acly* and *Cpt1a*. 84% KD efficiency for *Acly* and 71% KD efficiency for *Cpt1a*. ($n = 6$, unpaired $t$-test). **i** PyMT MIC frequency after treatment with vehicle or fluoroacetate ± SB204990. ($n = 5$[a], paired $t$-test). **j** Lung metastases in recipient wt FVB/N mice after TVI of PyMT cells treated with vehicle or fluoroacetate ± SB204990. ($n \geq 9$, unpaired $t$-test). **k** PyMT MIC frequency after treatment with acetate. ($n = 6$[a], paired $t$-test). **l** Lung metastases in recipient wt FVB/N mice after TVI of PyMT cells treated with vehicle or acetate. Left: quantification. Right: representative images of lung metastases of recipient mice. ($n = 7$, unpaired $t$-test). Paired $t$-test were two-tailed and by ratio. Unpaired $t$-test were parametric and two-tailed. Values shown correspond to means ± SEM. Source data are provided in the source data file. [a] signifies number of independent experiments or tumors.

but also MIC frequency, endogenous metastatic potential, and the expression of EMT-related genes in breast cancer cells.

It has been shown recently that for EMT to be activated, only few histone marks are required to be altered, and these epigenetic changes occur quickly—within the first 6 h of EMT induction[15]. The fact that we observe global differences in histone acetylation that are in the order of percentage change rather than in the order of several fold changes, is therefore, expected. In addition, cells exhibiting a partial EMT phenotype, where they oscillate between mesenchymal and epithelial features, are most efficient at metastasizing due to their plasticity[41], increased propensity to migrate and seed in clusters[42], and their ability to re-epithelialize upon colonization of the target tissue[43]. Since our RNA-sequencing revealed partial EMT features in the endogenous MICs, it is likely that metabolism serves as a mechanism for MICs to precisely modulate and fine tune the extent of both epithelial and mesenchymal characteristics needed during the different steps of metastatic dissemination. Other TCA cycle metabolites have been shown to regulate histone methylation of EMT genes, such as succinate[44] and fumarate[45]. However, methylation is a relatively stable epigenetic modification[46]. Since acetylation turns over in the timescale of seconds to minutes, highly plastic MICs may exploit acetyl-CoA generating pathways to enable this rapid phenotypic switching.

We further show that sorting for cells with high lipid storage is sufficient to identify MICs, and this high level of lipid stores is enriched in early micrometastatic lesions. We also observed that MICs have a heightened flux of lipids through these stores and pharmacologically disrupting this flux inhibits metastasis. This suggests that lipid stores may act as buffers to maintain FAO when lipid uptake is insufficient in order to ensure a continuous supply of acetyl-CoA for EMT-like phenotypic switching. In addition, lipid droplets have been shown to limit reactive oxygen species by sequestering polyunsaturated fatty acids that are vulnerable to chain peroxidation reactions[47]. Considering the existing evidence that detachment from substrate causes increased ROS due to reduced antioxidant generation[19,48], circulating tumor cells could also rely on lipid droplets to prevent peroxidation reactions by sequestering "excess" lipids and to battle heightened ROS through subsequent lipolysis and FAO for antioxidant (NADPH) generation.

We've now described several metabolic differences between MICs and non-MICs. What remains unknown is how these phenotypes arise and how they are maintained. Using an intracellular glucose biosensor, one study showed that intratumor metabolic heterogeneity can be maintained at least over one cell division[49]. It has also been shown that claudin-low, basal, and luminal breast cancer subtypes maintain the metabolic phenotype of their respective cells-of-origin (normal basal, luminal progenitor, and mature luminal mammary epithelial cells, respectively)[50]. These examples suggest that metabolic state is not only heritable but is also intrinsically regulated.

In this study, we show for the first time that acetate can induce EMT features, MIC frequency, and metastatic potential in breast cancer cells. Additionally, we demonstrate that the metastasis-promoting function of acetate can be blunted not just by blocking the direct enzymes that generate acetyl-CoA from acetate, ACSS1 and ACSS2[51],

but also by inhibiting FAO, ACLY, or readers of H3K27ac. Acetate is the most common short chain fatty acid produced by the gut microbiome and about 10% of human energy is estimated to stem from acetate metabolism[52]. While its concentration in the blood is between 50 to 200 μM, acetate concentration is known to increase 2 or 3-fold in response to dietary changes such as alcohol consumption (since alcohol is detoxified to acetate through alcohol dehydrogenases)[53] and high fat diet (HFD; through HFD-induced overrepresentation of certain gut microbiota)[54]. As it has been reported that both alcohol consumption[55] and HFD[56] are associated with increased metastatic potential in breast cancer, it appears possible that acetate-induced histone alterations, at least in part, play a mechanistic role in these settings, and may therefore be a suitable therapeutic target for certain patients.

Finally, our work demonstrates that precise regulation of histone acetylation is crucial for metastasis. Others have shown how histone acetylation is also important in other cancer promoting phenotypes such as in DNA damage repair[57] and resistance to conventional chemotherapy[58], indicating that perturbing histone acetylation can be an effective and multimodal treatment strategy. Pan-histone deacetylase (HDAC) inhibitors have been used to increase histone acetylation, however, while these can cause cancer cell death in cells with very high histone acetylation levels, this may also cause an undesired conversion of non-MICs to MICs. We have also shown that due to the plurality of mechanisms for acetyl-CoA generation and the inherent plasticity of MICs, blocking one pathway may not always be sufficient due to compensatory mechanisms such as the upregulation of ACSS2 in the absence of ACLY[33,59]. It may therefore be essential to target multiple acetyl-CoA generating pathways and additionally target downstream epigenetic modifiers coupled to closely monitoring response to treatment for full and long-lasting efficacy.

## Methods

### Mice

All animal procedures were performed in accordance with the Swiss Animal Welfare Regulations and had been approved by the Cantonal Veterinary Service of Canton Vaud. MMTV-PyMT mice in the FVB/N background and the rag2$^{-/-}$γc$^{-/-}$ mice in the NMRI background were obtained from JAX and were described previously[60,61]. DEREG mice in the BALB/c background were used as GFP-tolerant recipients and were obtained by backcrossing DEREG mice in the C57BL/6J background[62] to wildtype BALB/c mice for at least 10 generations. Wildtype FVB mice were bred inhouse, and wildtype BALB/c mice were purchased from Charles River Laboratories.

### Cell lines

The murine cancer cell line 4T1 was obtained from ATCC, the human cancer cell lines MDA-MB-231 and BT549 were a kind gift from Prof. Paloma Ordóñez Morán (University of Nottingham), and the human cancer cell lines, BT20 and HCC1806, were a kind gift from Prof. Cathrin Brisken (EPFL). Cell lines were maintained in Dulbecco's Modified Eagle Medium (Gibco #31966021) supplemented with 10%

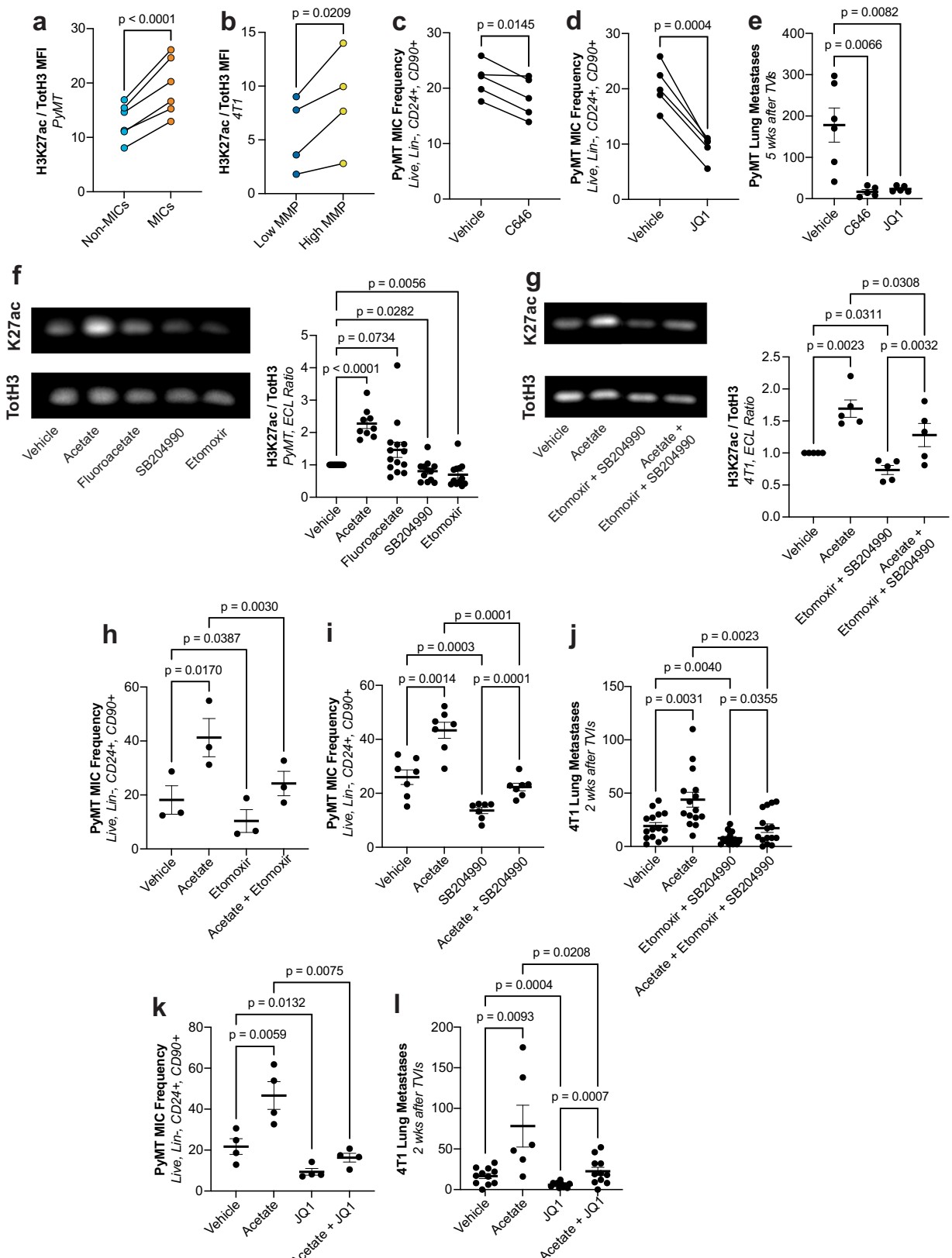

fetal bovine serum (FBS; Sigma-Aldrich), 20 mM HEPES buffered to pH 7.4 (HEPES), and penicillin-streptomycin (pen/strep; Gibco #15140122).

**Tumor cell isolation from spontaneous PyMT tumors**

MMTV-PyMT mice were sacrificed at -12–18 weeks of age for tumor cell isolation. Tumors were minced then digested with 37.5 μg/ml Liberase

TH (Roche #5401151001), 37.5 μg/ml Liberase TM (Roche #5401127001), and 25 μg/ml DNAse (Roche #11284932001) in DMEM/F12 (Gibco #31331028) supplemented with HEPES and pen/strep for 1 h at 37 °C. Cells were washed once in 4 °C 1x PBS supplemented with 0.5% BSA and 2 mM EDTA, then were passed through a 70 μm strainer, then washed an additional two times in DMEM/F12 supplemented with

**Fig. 4 | Metabolism supports histone acetylation to facilitate metastatic capacity in MICs. a** H3K27ac levels normalized to total histone 3 (TotH3) measured by flow cytometry in ex vivo PyMT non-MICs vs. MICs. ($n = 6^a$, paired *t*-test). **b** H3K27ac levels normalized to total histone 3 (TotH3) measured by flow cytometry in 4T1 cells sorted for high vs. low MMP. ($n = 4^a$, paired *t*-test). **c** MIC frequency in PyMT cells after treatment with vehicle or C646. ($n = 5^a$, paired *t*-test). **d** MIC frequency in PyMT cells after treatment with vehicle or JQ1. ($n = 5^a$, paired *t*-test). **e** Lung metastases in recipient wt FVB/N mice after TVI of PyMT cells treated with JQ1 or C646. ($n \geq 5$, unpaired *t*-test). **f** Left: representative Western blot from PyMT cells treated with vehicle, acetate, fluoroacetate, SB204990, or etomoxir. Right: quantification of ratio of H3K27ac (K27ac) signal vs. total histone 3 (TotH3) signal, expressed as the fold change over vehicle treated cells. ($n \geq 9^a$, paired *t*-test). **g** Left: representative Western blot from 4T1 cells treated with vehicle, acetate, etomoxir + SB204990, or acetate + etomoxir + SB204990. Right: quantification of ratio of H3K27ac (K27ac) signal vs. total histone 3 (TotH3) signal, expressed as the fold change over vehicle treated cells. ($n = 5^a$, paired *t*-test). **h** MIC frequency in PyMT cells after treatment with vehicle or acetate ± etomoxir. ($n = 3^a$, paired *t*-test). **i** MIC frequency in PyMT cells after treatment with vehicle or acetate ± SB204990. ($n = 7^a$, paired *t*-test). **j** Lung metastases in recipient wt Balb/c mice after TVI of 4T1 cells after treatment with vehicle, acetate, etomoxir + SB204990, or acetate + etomoxir + SB204990. ($n = 15$, unpaired *t*-test). **k** MIC frequency in PyMT cells after treatment with vehicle or acetate ± JQ1. ($n = 4^a$, paired *t*-test). **l** Lung metastases in recipient wt Balb/c mice after TVI of 4T1 cells after treatment with vehicle or acetate ± JQ1. ($n \geq 6$, paired *t*-test). Paired *t*-test were two-tailed and by ratio. Unpaired *t*-test were parametric and two-tailed. Values shown correspond to means ± SEM. Source data are provided in the source data file. $^a$ signifies number of independent experiments or tumors.

HEPES and pen/strep. At this point, cells were stained for flow cytometry or lysed for RNA isolation. Otherwise, cells were seeded onto collagen-coated plates with DMEM/F12 supplemented with 2% FBS, HEPES, pen/strep, 20 ng/ml hEGF (Gibco #PHG0313), and 10 µg/ml hInsulin (Sigma-Aldrich #I9278-5ML).

## In vitro treatments

**Galactose supplementation.** For PyMT cells, after overnight seeding in complete media as outlined above, media was changed to glucose- and glutamine-free DMEM (Gibco #A1443001) supplemented with 2% FBS, HEPES, pen/strep, 1x GlutaMAX (Gibco # 35050061), hEGF, hInsulin, 0.5x MEM Non-Essential Amino Acids Solution, and 20 mM galactose or 20 mM glucose. Cells were incubated in this medium for 36–72 h prior to trypsinization for flow cytometry. For 4T1 cells, after overnight seeding in complete media as outlined above, media was changed to glucose- and glutamine-free DMEM (Gibco #A1443001) supplemented with 10% FBS, HEPES, pen/strep, 1x GlutaMAX (Gibco # 35050061), and 20 mM galactose or 20 mM glucose. Cells were incubated in this medium for 48 h prior to trypsinization for TVIs.

**Inhibitor treatments.** After overnight seeding, media was changed to media containing inhibitors of interest the next day. PyMT cells were incubated with the treatments for 72 h and 4T1 cells and human cell lines were incubated with the treatments for 48 h. The list of inhibitors, including their catalog numbers and the concentrations at which they were used for each cell line can be found in Table 1.

## Flow cytometry

**PyMT CSC frequency.** Biotin-conjugated antibodies used for lineage depletion are Ter119 (Biolegend #116203, clone Ter119), CD31 (Biolegend #102503, clone Mec13.3), and CD45 (Biolegend #103104, clone 30-F11). PyMT CSCs are then identified by staining for CD24 (Biolegend #101807, clone MI/69) and CD90.1 (Biolegend #202526, clone OX-7).

**ALDEFLUOR assay.** Staining was performed according to the manufacturer's instructions (STEMCELL Technologies #01700). Briefly, cells were incubated with ALDEFLUOR reagent with or without DEAB for 30 min at 37 °C. Cells were washed with the provided assay buffer and kept on ice during acquisition.

**Mitochondrial probes.** Cells were stained with 20 nM MitoSpy Red CMXRos (Biolegend #424801) in media for mitochondrial membrane potential in live cells and 200 nM in fixed cells at 37 °C. Cells were stained with 50 nM MitoSpy Green FM (Biolegend #424806) for mitochondrial mass in live cells at 37 °C.

**Lipid storage.** Cells were stained according to a previously published protocol[63]. In brief, cells were washed twice with 1x PBS then were stained with 2 µm BODIPY 493/503 (Invitrogen #D3922) in

1x PBS for 15 min at 37 °C. Cells were no longer subjected to FBS after staining.

**Lipid uptake.** Cells were stained according to a previously published protocol[64]. In brief, attached cells were incubated with 37 °C 2 µm BODIPY FL C12 (Invitrogen #D3822) in 0.1% Fatty-Acid Free BSA (Sigma-Aldrich #126609) in 1x HBSS (Gibco #14175095) for 30 s to 5 min. Staining was quenched by addition of 4 °C 0.2% BSA in 1x PBS.

**Histone marks.** Cells in suspension were first stained for viability using LIVE/DEAD Fixable Dead Cell Stain Kit (Invitrogen #L23105). Cells were then fixed and permeabilized using the Foxp3/Transcription Factor Staining Kit (eBioscience #00-5523-00) according to the manufacturer's protocol. Cells were then stained with antibodies against H3K27ac (CST #8173, clone D5E4) or Total H3 (Novus Biologicals #NBP2-59277, clone 1B1B2) at room temperature for 1 h, then stained with anti-rabbit (Thermo Fisher Scientific #A-21206) or anti-mouse (Thermo Fisher Scientific #A-11029) secondary antibody at room temperature for 1 h.

**Cell death.** Cells were stained for Annexin V (Biolegend #640905) according to manufacturer's protocol and DAPI. Living cells were considered as Annexin V negative and DAPI negative.

**Proliferation.** Cells were stained using CellTrace Violet Proliferation Kit (Invitrogen #C34557) prior to plating and were measured 36 h later. If not otherwise specified, viability was measured by DAPI staining.

Measurements were analyzed using the FlowJo software v10.7.2.

## Seahorse assays

**Tumorsphere generation.** Single cells were seeded into polyHEMA coated plates[65] with DMEM/F12 supplemented with 2% FBS, HEPES, pen/strep, 20 ng/ml hEGF, 10 µg/ml hInsulin, and 4 ng/ml of Heparin. Cells were cultured for 96 h prior to assay.

**Mito stress test.** Cells were grown on Seahorse XF96 cell culture dishes for a minimum of 4 h prior to media change. For the assay, cells were maintained in 1x KHB media supplemented with pen/strep, 20 ng/ml hEGF, 10 µg/ml hInsulin, and 0.5 mM L-Carnitine (Sigma-Aldrich #C0158), as described in the manufacturer's protocol. OCR and ECAR were measured every 6 min, and treatments were administered every 30 min. Glucose to 10 mM final concentration was injected first, followed by oligomycin (Sigma-Aldrich #495455-10MG) to 1 µM final concentration, then FCCP (Sigma-Aldrich #C2920-10MG) to 2 µM final concentration, then finally Rotenone (Sigma-Aldrich #R8875-1G) and Antimycin A (Sigma-Aldrich #A8674-25MG) to a final concentration of 0.5 µM each.

**Mito fuel flex test.** Cells were grown on Seahorse XF96 cell culture dishes for a minimum of 4 h prior to media change. For the assay, cells

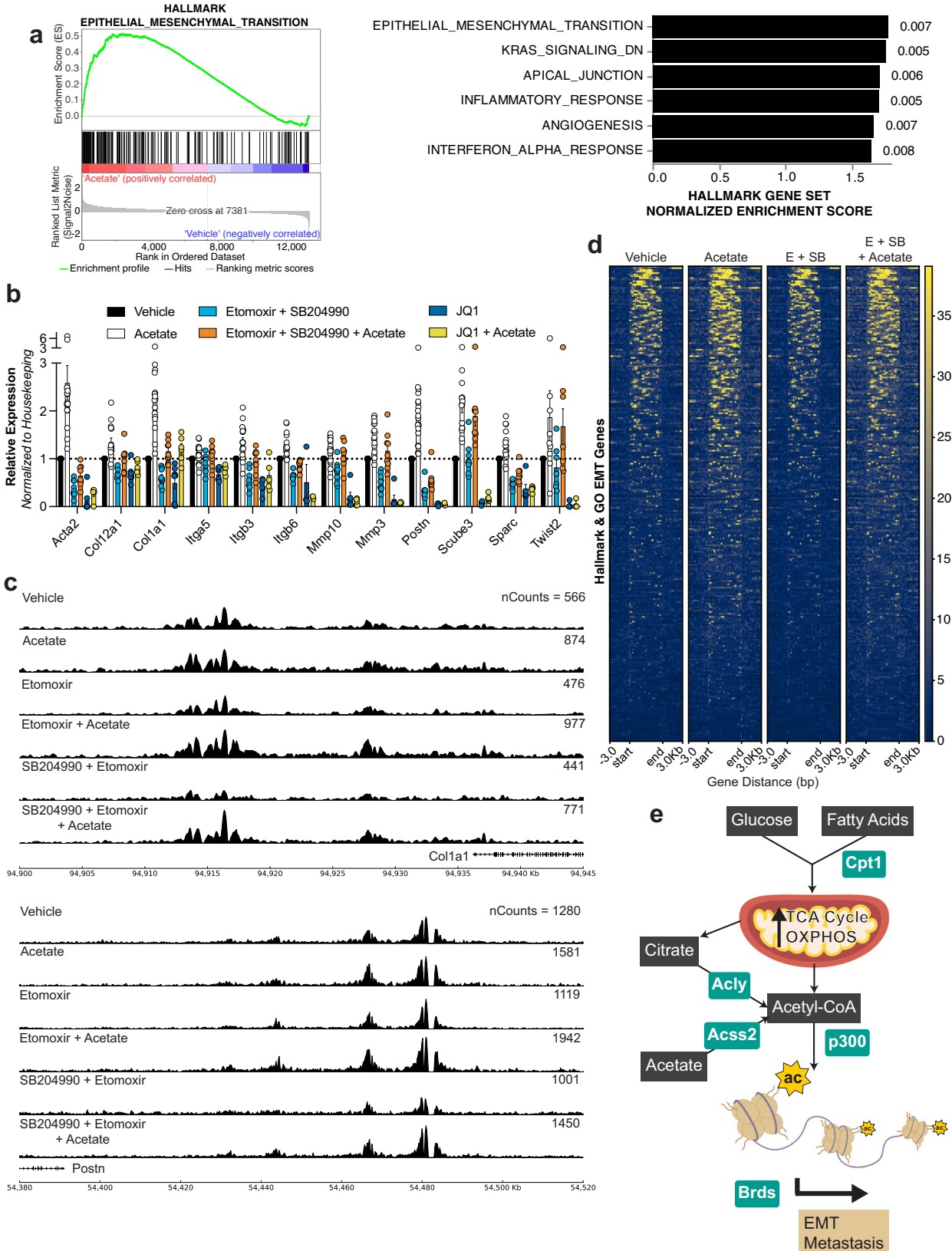

were maintained in DMEM/F12 supplemented with 2% FBS, pen/strep, and HEPES. OCR was measured every 6 min, and treatments were administered every 30 min. Final concentrations for drug injections were 40 μM for etomoxir (Sigma-Aldrich #E1905-5MG), 10 μM for BPTES (Sigma-Aldrich #SML0601-5MG), and 20 μM for UK5099 (Sigma-Aldrich #PZ0160-5MG). Fuel dependence was calculated by the difference between the area under the curve (AUC) of basal OCR (timepoints 1–5) and after single inhibitor injection (timepoints 6–10) over the difference between the AUC of basal OCR and after double inhibitor injection (timepoints 11–15). Fuel capacity was calculated by the difference between the area under the curve (AUC) of basal OCR (timepoints 1–5) and after double inhibitor injection (timepoints 6–10)

**Fig. 5 | Metabolism-induced histone acetylation enables expression of EMT-related genes. a** Left: gene set enrichment plot for the HALLMARK_EPITHE-LIAL_MESENCHYMAL_TRANSITION signature in acetate- vs. vehicle-treated 4T1. Right: top upregulated gene sets from the Hallmark Collection in the molecular signatures database (mSigDB) in acetate- vs. vehicle-treated 4T1 cells. Numbers beside the bars are the false discovery rates. (*n* = 4[a]). **b** Relative expression of EMT-related genes in 4T1 cells in response to the indicated treatments normalized to housekeeping genes and plotted as a fold change against vehicle-treated. (*n* ≥ 3[a]). **c** H3K27ac ChIP-Seq tracks for *Col1a1* (top) and *Postn* (bottom) from 4T1 cells in response to the indicated treatments. Numbers on the right-hand side of the tracks indicate the associated scaled counts per gene. **d** Heatmap of identified H3K27ac chip-seq signals on the regions associated with the EMT genes found in union of the

HALLMARK_EPITHELIAL_MESENCHYMAL_TRANSITION and GOBP_EPITHE-LIAL_TO_MESENCHYMAL_TRANSITION gene sets upon treatment of 4T1 cells with vehicle, acetate, etomoxir (E) + SB204990 (SB), and acetate + E + SB. **e** Schematic of the working model proposed. Fuels such as glucose and fatty acids are pre-ferentially utilized in the mitochondria of MICs to generate citrate and acetyl-CoA. Other pathways such as acetate to acetyl-CoA conversion contribute to this pool. Dependence on acetyl-CoA generating pathways is crucial to facilitate the acet-ylation of histones, particularly on H3K27, for the expression of EMT-related genes and subsequent metastatic capacity. Values shown correspond to means ± SEM. Source data are provided in the source data file. [a] signifies number of independent experiments or tumors.

over the difference between the AUC of basal OCR and after single inhibitor injection (timepoints 11–15).

### Proliferation assay
Cells were treated for 48 to 72 h prior to trypsinization and plating for proliferation. If KDs were used, sufficient KD was first confirmed prior to plating for proliferation. For each measurement, MTT reagent was added to a final concentration of 0.5 mg/ml for 3.5 to 4 h. Media was aspirated from the plates, and MTT solvent (Isopropanol + 4 mM HCl + 0.1% NP40) was added. After 5 min on a benchtop shaker, the absorbance was read using a spectrophotometer (TECAN) at 590 nm wavelength.

### Metabolomics measurements
Adherent cells were washed twice in 1x PBS prior to snap freezing in liquid nitrogen. Cells were scraped in a 2:2:1 ratio of methanol, acet-onitrile, and water supplemented with 0.1 M formic acid then homo-genized in the Cryolys Precellys 24 sample Homogenizer (Bertin Technologies, Rockville, MD, US) with ceramic beads. Homogenized extracts were centrifuged and the resulting supernatant was collected and dried in a vacuum concentrator (LabConco, Missouri, US). Dried sample extracts were resuspended with 80% MeOH containing the internal standards and injected into the HILIC-MS/MS system. Extrac-ted samples were analyzed by Hydrophilic Interaction Liquid Chro-matography coupled to tandem mass spectrometry (HILIC−MS/MS)[66,67] in negative ionization mode using a 6495 triple quadrupole system (QqQ) interfaced with 1290 UHPLC system (Agilent Technol-ogies). Chromatographic separation was carried out on a SeQuant ZIC-pHILIC column (100 mm, 2.1 mm I.D. and 5 µm particle size, Merck, Damstadt, Germany). Raw LC-MS/MS data was processed using the Agilent Quantitative analysis software (version B.07.00, MassHunter Agilent technologies). For absolute quantification, calibration curves and the stable isotope-labeled internal standards (IS) were used to determine the response factor. Linearity of the standard curves was evaluated for each metabolite using a 10-point range; in addition, peak area integration was manually curated and corrected when necessary. Concentrations were normalized to protein content measured by BCA assay (Thermo Scientific #23225) following the metabolite extraction.

### Western blotting
Cells were washed twice in 1x PBS supplemented with 5 mM sodium butyrate, then lysed with 1x PBS supplemented with 1% Triton X, 5 mM sodium butyrate, anti-protease inhibitors (100 µM AEBSF, 10 µM bes-tatin, 10 µM E64, 10 µM leupeptin, 10 µM pepstatin A, 1 mM DTT), and anti-phosphotatse inhibitors (2.5 mM sodium pyrophosphate, 1 mM ß-glycerophosphate, 1 mM sodium orthovanadate, 2.5 mM sodium fluoride). Histones were precipitated using 0.2 N HCl overnight then quantified by BCA assay. Proteins were separated by electrophoresis, transferred to PVDF membranes, then blocked with 5% BSA prior to overnight incubation with primary antibodies (H3K27ac: CST #8173 clone D5E4, Total H3: CST #3638 clone 96C10) at 4 °C.

Immunoreactive bands were visualized using HRP-conjugated sec-ondary antibodies (Anti-Rabbit: Jackson Immunosearch #711-035-152, Anti-Mouse: Promega #W4021) and the detection reagent Wester-nBright Sirius HRP substrate (Advansta #K-12043-D10). For protein normalization, PVDF membranes were stripped with 2% SDS, 62.5 mM Tris HCl pH6.8, and 0.8% ß-mercaptoethanol, washed extensively, then reprobed.

### RNA isolation and sequencing
Primary PyMT cells were plated overnight, then sorted the next day for Lin⁻CD24⁺CD90⁺ and Lin⁻CD24⁺CD90⁻ cells. Cells were directly col-lected into lysis buffer, and RNA isolation was subsequently performed with the Qiagen RNeasy Mini Kit (Qiagen #74104) according to the manufacturer's protocol. RNA quality was assessed using the Agilent DNF-472T33 HS RNA (15 nt) Kit (Agilent Technologies) on fragment analyzer (Qiaxcel). Library preparation was performed using the Tru-Seq mRNA stranded method (Illumina), then quality was checked using TapeStation TS4200 (Agilent). Libraries were then sequenced on a NextSeq500 sequencer (Illumina) for 75 cycles at 40 million reads per sample. Raw sequencing reads were aligned against the mouse tran-scriptome (GRCm38/mm10 release) and the quantification was done at the gene level using the Bowtie2/samtools pipeline. This data is deposited in GEO and can be accessed under GSE236222.

4T1 cells were plated overnight, then treated for 48 h. Cells were harvested by trypsinization, and the cell pellet was resuspended in lysis buffer. For sorted 4T1 cells, cells were stained for 30 min with 20 nM MitoSpy Red CMXRos, trypsinized, sorted for high and low MMP (5% each population), spun down, and resuspended in lysis buffer. RNA isolation was then performed using the High Pure RNA Isolation Kit (Roche #11828665001) according to the manufacturer's protocol. RNA Quality was assessed by TapeStation TS4200, then libraries were prepared using the Illumina Stranded mRNA Prep Kit. Library quality was then assessed by TapeStation TS4200, then sequenced on the NextSeq500 (Illumina) for 75 cycles at 40 million reads per sample. Raw sequencing reads were aligned against the mouse transcriptome (GRCm38/mm10 release) and the quantification was done at the gene level using STAR[68]. These data are deposited in GEO and can be accessed under GSE220991 and GSE236185.

**Gene set enrichment analysis.** Count matrices were normalized to total library size, filtered for genes expressed by at least half of the samples, then were subjected to unbiased gene set enrichment ana-lysis using the GSEA software v4.2.3[69–72]. Analysis was performed against the Hallmark Gene Set collection and the Gene Ontology Gene Set collection with 1000 gene set permutations. Gene set enrichment was considered significant after Signal2Noise gene ranking at a false discovery rate under 0.05 for sets containing at least 3 replicates per condition (PyMT MICs vs. non-MICs and 4T1 cells treated with vehicle vs. acetate). For the high vs. low MMP dataset, gene set enrichment was considered significant at a false discovery rate under 0.25 after log2 Ratio of Classes gene ranking.

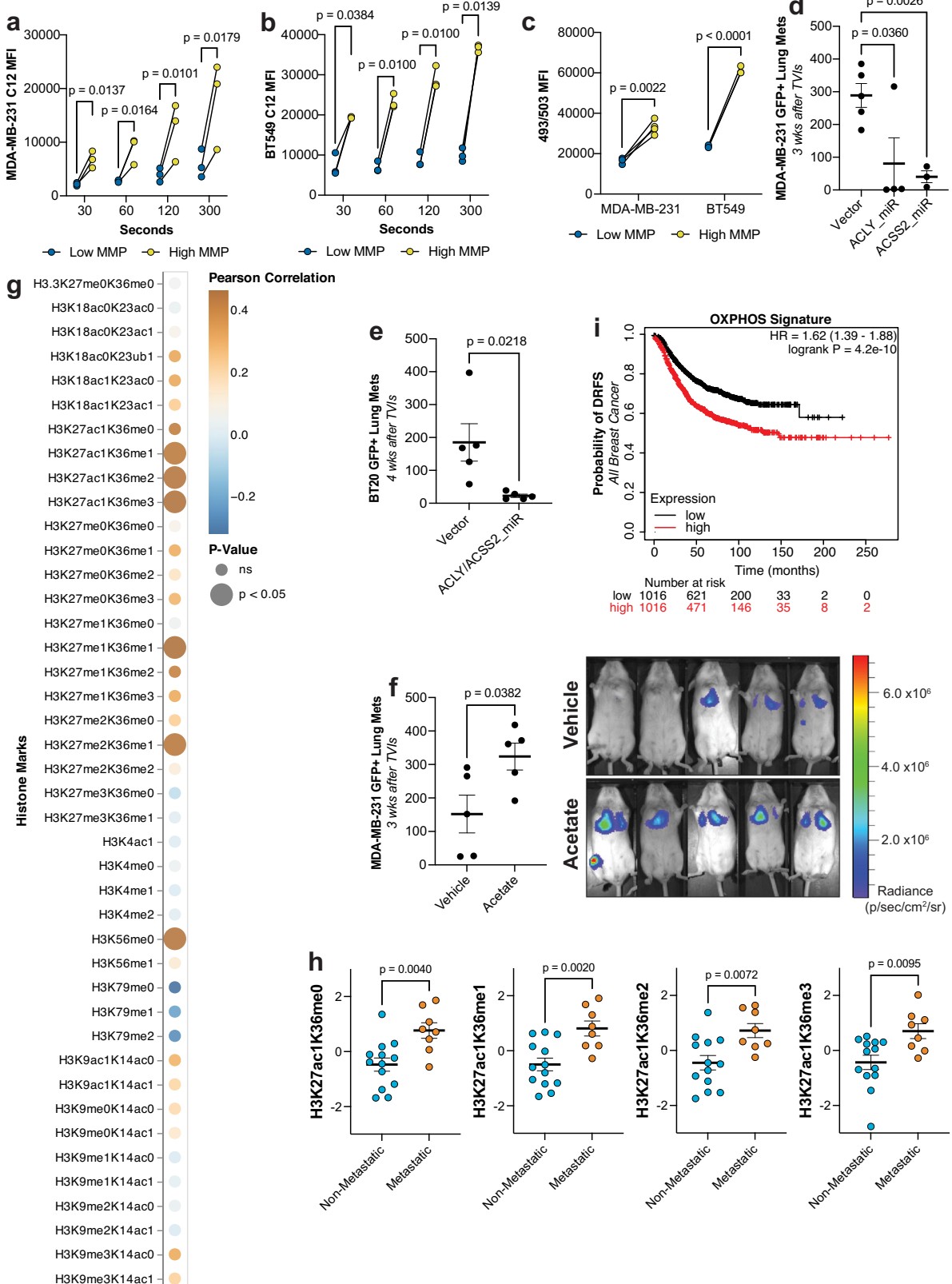

**Real time qPCR**
Complementary DNA was synthesized using the Maxima H Minus Reverse Transcriptase (Thermo Scientific #EP0752) and oligodT priming. Quantitative PCR was performed using the PowerUp SYBR Green Master Mix (Applied Biosystems #A25741) on a QuantStudio 6 Flex Real Time PCR Machine (Applied Biosystems). Primer sequences were either taken from the PrimerBank database[73–75] or were designed in-house and can be found listed in Table 2.

**Lentivirus production for miR-mediated KDs**
Lentiviruses were generated in 293T cells using 3rd generation packaging vectors. Viral particles were concentrated by

**Fig. 6 | Metabolic dependencies for mitochondrial activity and acetyl-CoA define metastatic activity in human breast cancer cell lines. a** LCFA uptake measured by MFI of MDA-MB-231 cells sorted for low and high MMP after exposure to BODIPY FL C12 for specified durations. (*n* = 3ᵃ, paired *t*-test). **b** LCFA uptake measured by MFI of BT549 cells sorted for low and high MMP after exposure to BODIPY FL C12 for specified durations. (*n* = 3ᵃ, paired *t*-test). **c** Lipid stores measured by MFI of MDA-MB-231 and BT549 cells sorted for low and high MMP after staining with BODIPY 493/503. (*n* = 4ᵃ, *n* = 3ᵃ, respectively, paired *t*-test). **d** Number of GFP+ metastases in recipient Rag2⁻/⁻γc⁻/⁻ mice after TVI of MDA-MB-231 cells expressing a vector control or miR-mediated KDs of *ACLY* or *ACSS2*. 80% and 75% KD efficiency, respectively. (*n* = 5, *n* = 4, *n* = 3, respectively, unpaired *t*-test). **e** Number of GFP+ metastases in recipient Rag2⁻/⁻γc⁻/⁻ mice after TVI of BT20 cells expressing a vector control or miR-mediated KDs of *ACLY* and *ACSS2*. 78% and 73% KD efficiency, respectively. (*n* = 5, unpaired *t*-test). **f** Number of GFP+ lung

metastases in recipient Rag2⁻/⁻γc⁻/⁻ mice after TVI of MDA-MB-231 cells treated with vehicle or acetate. Right: bioluminescence images of mice at day 21 after TVI. (*n* = 5, unpaired *t*-test). **g** Heatmap of Pearson scores and two-tailed *p* values calculated by correlating the metastatic potential of 21 TNBC cell lines derived from the MetMap study to their corresponding histone profile generated by the CCLE. **h** Scaled quantification of selected histone marks from (**g**) between metastatic and non-metastatic TNBC cell lines. (*n* = 8, *n* = 13, respectively, unpaired *t*-test). **i** Kaplan−Meier plots depicting distant relapse-free survival (DRFS) of breast cancer patients deriving from the microarray cohort of KM plotter that have been stratified by the mean expression of OXPHOS-related genes and segregated based on the median cut-off. MFI mean fluorescence intensity. Paired *t*-test were two-tailed and by ratio. Unpaired *t*-test were parametric and two-tailed. Values shown correspond to means ± SEM. Source data are provided in the source data file. ᵃ signifies number of independent experiments or tumors.

ultracentrifugation for 2 h and 30 min at 22,000 rpm. Titration was performed in 293Ts. Cells were then infected overnight with lentiviruses in the presence of 8 μg/ml polybrene. MicroRNA sequences for KDs are listed in Table 3.

### ChIP-seq

Chromatin immunoprecipitation was performed using the SimpleChIP Enzymatic Chromatin IP Kit (CST #9003) according to the manufacturer's protocol. Briefly, adherent cells were washed then fixed in 1x PBS supplemented with 1% methanol-free PFA (Thermo Scientific #28908). The reaction was quenched with 1X glycine, cells were pelleted, then stored at −80 °C. Cells were lysed, and chromatin was sheared by adding 0.5 ul Micrococcal nuclease for 20 min at 37 °C. Nuclei were then lysed by sonication. After addition of 7.5% of human spike-in, sheared mouse chromatin was incubated overnight at 4 °C with 1:100 H3K27ac antibody (CST #8173). Protein G magnetic beads were used to pull down antibody-bound fragments then eluted at 65 °C for 30 min. Reverse crosslinking was done with NaCl and Proteinase K then purified with columns.

Libraries were prepared using the NEBNext Ultra II DNA Prep Kit (NEB #E765S) according to the manufacturer's protocol then quality was assessed on the TapeStation TS4200. Libraries were then sequenced on NovaSeq6000 in a PE60 run with an average sequencing depth of 40 million reads per sample with Illumina protocol #1000000106351 v03.

Raw sequences were pre-processed and aligned to a merged GRCh38 and GRCmm10 reference genome through the nf-core/chipseq v1.2.2 workflow[76]. Scaling factors for the mouse library were computed by normalizing to the percentage of human spike-in then normalized to the highest scaling factor. BigWig files were generated using the BamCoverage tool from deepTools v3.5.1 then tracks were plotted using pyGenomeTracks v3.7[77]. The heatmap was generated

using the computeMatrix (scaled mode; gene regions were scaled to 5000 bp) and plotHeatmap function from deepTools. The plotted EMT genes comes from the union of genes present in HALLMARK_EPITHELIAL_MESENCHYMAL_TRANSITION and GOBP_EPITHELIAL_TO_MESENCHYMAL_TRANSITION available from the Molecular Signature Database. The BED file can be found in Supplementary Data 1. The data is deposited in GEO and can be accessed under GSE236223.

### Lipid droplet measurements

Immunostaining on fresh frozen tissue was performed on 5 μm sections after 1 h fixation with 4% paraformaldehyde and a 10-min permeabilization with 0.1% Triton X-100. Primary antibody staining was done for 1 h. Sections were then stained for 30 min with Oil Red O then washed extensively with water. Nuclei were stained with DAPI, then mounted using Fluoromount G (SouthernBiotech #0100-01).

Images were analyzed by QuPath v0.4.2. Areas containing tumor cells were selected manually by annotating areas that were GFP positive and had enlarged nuclei. Lipid droplets and nuclei were detected by setting a size and intensity threshold using the pre-trained 2D_versatile_fluo 2D_paper_dsb2018 model from StarDist[78,79].

### Experimental metastasis assay

Tumor cells were treated with inhibitors or supplements for 48 h for 4T1, BT20, and MDA-MB-231 or 72 h for PyMT cells. Cells were trypsinized, washed three times in cold 1x HBSS, then resuspended in 100 μl of 1x HBSS for TVIs. 5 × 10⁵ PyMT cells were injected into each wildtype FVB/N recipient mice, 5 × 10⁴ 4T1 cells were injected into each wildtype Balb/c, DEREG, or Rag2⁻/⁻γc⁻/⁻ recipient mice, 3 × 10⁵ BT20 cells were injected into each Rag2⁻/⁻γc⁻/⁻ mice, and 5 × 10⁴ MDA-MB-231 cells were injected into each Rag2⁻/⁻γc⁻/⁻ mice. Recipient mice were all around 8–20 weeks of age. Mice were euthanized at a fixed time point−Day 35 for PyMT, Day 14 for 4T1 (Day 8 if injected into Rag2⁻/⁻γc⁻/⁻ recipients), and Day 28 for BT20, and MDA-MB-231. Lung lobes were separated and macrometastases were counted using a brightfield stereomicroscope. If cells were expressing GFP, GFP+ nodules were counted by flattening each lung lobe between two microscope slides then taking images using a fluorescent stereomicroscope (Nikon SMZ18 stereomicroscope). Images were then analyzed using Fiji v2.9.0 by size and intensity threshold and using the "analyze particles" function. All animal procedures were in accordance with Swiss legislation on animal experimentation.

### Orthotopic experiments

3.5 × 10⁵ PyMT cells were resuspended in 30 μl of 2 mg/ml collagen I (Gibco #A1048301) and injected into the fourth mammary pad of wildtype FVB/N recipient mice (age between 8–20 weeks). Tumor size was determined by caliper measurements and calculated using the formula: $V = (D \cdot d^2)/2$, where $d$ is the shorter and $D$ is the longer tumor axis. Tumors were resected just before reaching 1 cm³. Thirty

### Table 1 | List of inhibitors used along with their respective catalog numbers and the concentrations at which they are used for the two mouse models

| Treatment | Catalog number | PyMT | 4T1 |
|---|---|---|---|
| Oligomycin | Sigma-Aldrich #75351 | 1 μM | 1 μM |
| Acetate | Sigma-Aldrich # S5636-250G | 5 mM | 5 mM |
| Fluoroacetate | Toronto Research Chemicals #S960143 | 5 mM | 5 mM |
| Etomoxir | Sigma-Aldrich #E1905 | 40 μM | 50 μM |
| ATGListatin | SAGECHEM Limited | 20 μM | 40 μM |
| T863 | Cayman Chemical #25807 | 10 μM | 20 μM |
| PF-06424439 | Cayman Chemical #17680 | 20 μM | 40 μM |
| SB204990 | Tocris Bioscience #4962 | 30 μM | 50 μM |
| JQ1 | Sigma-Aldrich #SML0974 | 50 nM | 500 nM |
| C646 | Cayman Chemical #10549 | 1 μM | n/a |

**Table 2 | List of primers used to RT-qPCR genes of interest**

| Target gene | MGH primer ID | Forward primer | Reverse primer |
|---|---|---|---|
| mHprt | n/a | TGTATACCTAATCATTATGCCGAGG | CAGGTCAGCAAAGAACTTATAGC |
| mRpl19 | n/a | CTGATCAAGGATGGGCTGAT | GGCAGTACCCTTCCTCTTCC |
| mRpl22 | 6677775a1 | AGCAGGTTTTGAAGTTCACCC | CAGCTTTCCCATTCACCTTGA |
| mYwhaz | 6756041a1 | GAAAAGTTCTTGATCCCCAATGC | TGTGACTGGTCCACAATTCCTT |
| mPolr2m | 30519919a1 | TTGCCCGACAAAGGTAAAAGA | TGTCTATTGAGGAACAGTCAGGA |
| mAcly | 18204829a1 | ACCCTTTCACTGGGGATCACA | GACAGGGATCAGGATTTCCTTG |
| mAcss1 | 18034773a1 | GTTTGGGACACTCCTTACCATAC | AGGCAGTTGACAGACACATTC |
| mAcss2 | 31980996a1 | AAACACGCTCAGGGAAAATCA | ACCGTAGATGTATCCCCCAGG |
| mCpt1a | n/a | TGCCTCTATGTGGTGTCCAA | ACAACCTCCATGGCTCAGAC |
| mCpt2 | 162138914c1 | CAGCACAGCATCGTACCCA | TCCCAATGCCGTTCTCAAAAT |
| mActa2 | 6671507a1 | GTCCCAGACATCAGGGAGTAA | TCGGATACTTCAGCGTCAGGA |
| mPostn | 7657429a1 | CCTGCCCTTATATGCTCTGCT | AAACATGGTCAATAGGCATCACT |
| mItga5 | n/a | ATGGCTCAGACATCCACTCC | GGTCATCTAGCCCATCTCCA |
| mItgb3 | 7949057a1 | CCACACGAGGCGTGAACTC | CTTCAGGTTACATCGGGGTGA |
| mItgb6 | 10946686a1 | CAACTATCGGCCAACTCATTGA | GCAGTTCTTCATAAGCGGAGAT |
| mCol1a1 | n/a | CTGCACGAGTCACACCGGAA | AGGCAGGGCCAATGTCTAGT |
| mCol12a1 | 6680960a1 | AAGTTGACCCACCTTCCGAC | GGTCCACTGTTATTCTGTAACCC |
| mMmp3 | 6754714a1 | ACATGGAGACTTTGTCCCTTTTG | TTGGCTGAGTGGTAGAGTCCC |
| mMmp10 | n/a | TTTTAAAGGAAGTCAGTTCTGGG | ATGTCTTGTCTCATCAAATCTCC |
| mScube3 | 52138545c1 | CATCTGCCAGAATACCCCACG | GTCCTCCCGTTCACACTCATC |
| mSparc | n/a | GCATCAAGGAGCAGGACATCAA | ACGGTGAGCTTATGCAACTCC |
| mTwist2 | n/a | CTACAGCAAGAAATCGAGCG | GGCCTCGTTGAGCGACT |
| hTBP | n/a | CGAAACGCCGAATATAATCCCA | GACTGTTCTTCACTCTTGGCTC |
| hACLY | 38569422c2 | ATCGGTTCAAGTATGCTCGGG | GACCAAGTTTTCCACGACGTT |
| hACSS2 | 334724454c1 | AAAGGAGCAACTACCAACATCTG | GCTGAACTGACACACTTGGAC |

**Table 3 | Sequences for the microRNAs used to knockdown genes of interest**

| Target gene | Sense | Anti-sense |
|---|---|---|
| GFP | CCAGCCACAACGTCTATATCA | TGATATAGACGTTGTGGCTGT |
| mCherry | TCGAGTTCATCTACAAGGTGA | TCACCTTGTAGATGAACTCGC |
| mAcly | GAAGCCAGACCAGTTAATCAA | TTGATTAACTGGTCTGGCTTG |
| mAcss1 | GACAGTGGTATTTGCTGGCTT | AAGCCAGCAAATACCACTGTG |
| mAcss2 | TCCAAGGAATTTTACTGGAAA | TTTCCAGTAAAATTCCTTGGC |
| mCpt1a | TCCGATCATGGTTAACAGCAA | TTGCTGTTAACCATGATCGGC |
| mCpt2 | GAGAGCTCAGACAGAAGTTGA | TCAACTTCTGTCTGAGCTCTG |
| hACLY | CAGTCACAATCTTTGTCCGAA | TTCGGACAAAGATTGTGACTT |
| hACSS2 | CAGCTTGGAGATAAAGTTGCT | AGCAACTTTATCTCCAAGCTT |

days after tumor resection, mice were euthanized, and the number of spontaneous lung metastases were counted. All animal procedures were in accordance with Swiss legislation on animal experimentation.

## CCLE dataset analysis

Chromatin profiling and metastasis potential datasets were downloaded from the CCLE depmap portal. Mean metastatic potential was calculated by averaging metastatic potential across the 5 different sites of metastatic colonization. Values were re-scaled to the TNBC subset using StandardScaler from the scikit-learn package (v1.2.1)[80]. Pearson coefficients and two tailed p values for the correlations between scaled abundances of histone marks and mean metastatic potential were calculated using the pearsonr function from the SciPy Stats package (v1.10.1; Supplementary Table 1)[81]. Plots were then generated using the mark_point function of Altair (v4.2.2)[82].

## Kaplan-Meier

Kaplan-Meier plots for relapse-free survival were generated using the breast cancer cohort (mRNA gene chip) or the TCGA cohort (mRNA RNA-seq, breast cancer patients from the pan-cancer cohort) using KM-plotter[83]. Patients were stratified based on either the median or quartile cut-offs. For the microarray cohort, JetSet best probe sets were used for the genes of interest. The OXPHOS signature was generated from the genes that were identified as enriched from the HALLMARK_OXIDATIVE_PHOSPHORYLATION and GOBP_OXIDATIVE_PHOSPHORYLATION by GSEA in the sorted PyMT MICs compared to the non-MICs (Supplementary Data 2). Mean expression of all genes was used as input to stratify the patients.

## Statistical analysis

Processing and data analysis of microscopy images were performed in Fiji or QuPath. Flow cytometry files were analyzed using FlowJo.

Experimental data are presented as mean ± SEM and statistical computations were performed using GraphPad Prism9 or SciPy stats. Statistical significance was assessed by two-tailed unpaired parametric $t$-test, two-tailed paired ratio $t$-test, or ordinary one-way ANOVA, data were considered significant at $p \le 0.05$.

## Reporting summary

Further information on research design is available in the Nature Portfolio Reporting Summary linked to this article.

## Data availability

Data from the CCLE were downloaded from the depmap portal: https://depmap.org/portal/download/all/, specifically the files labeled "CCLE_GlobalChromatinProfiling_20181130.csv" and "Supplementary Table 02 basal breast cancer cohort met potential.xlsx". Source data underlying each figure are provided with this paper. High-throughout data can be accessed under the GEO SuperSeries GSE220992 (RNA-seq data: GSE220991, GSE236185, and GSE236222; ChIP-Seq data: GSE236223). Source data are provided with this paper.

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

## Acknowledgements

C.M.Y., L.B., and A.M.A. were supported in part by grants from the SNF (513310, 514040, 514593) and the Swiss League against Cancer (532237, 531882) to J.H. We thank R.A. Lambuta, and N. Katanayeva for providing guidance on the ChIP preparation, B. Mangeat for help with the RNA-seq and ChIP-Seq, M. Norkin for analyzing the sequencing data, S. Rettie for computational support, and all the EPFL core facilities including the GECF, SIB, HCF, CPG, FCCF, and BIOP, Agora core facilities, including the AIVC, MPF, CIF, and FCF, and the UNIL MEP for their tremendous help during various parts of the project.

## Author contributions

C.M.Y.: conceptualization, methodology, investigation, formal analysis, writing—original draft, visualization; L.B.: investigation; P.D.: methodology, investigation; A.M.A.; investigation; A.S.M.: conceptualization, writing—review and editing; J.H.: conceptualization, supervision, funding acquisition, writing—review and editing.

## Competing interests

The authors declare no competing interests.
