## [Peer Review file · Nature Communications]

REVIEWER COMMENTS

Reviewer #1 (Remarks to the Author):

In the paper entitled “Metabolic dependencies of metastasis-initiating cells in breast cancer” Megan and colleagues investigate the role of mitochondrial function and the ensuing metabolic features of tumour metastasis. In particular, the authors found that cells with high mitochondrial potential exhibit higher metastatic ability and that metastasis depends on fatty acid oxidation (FAO), which in turn, is required for acetyl-CoA production. In addition, they report that acetyl-CoA is used to establish the H3K27ac mark on promoters of EMT-related genes, inducing their expression. Lastly, they conclude that by modulating either the acetyl-CoA-generating pathways or specific H3K27Ac writers/readers they can influence the tumorigenic potential.

Although aimed to address a relevant and timely question in the field, this study suffers from major conceptual and technical limitations that undermine the strength of the conclusions. First, the authors used the two different cell models interchangeably throughout the study and perform some experiments in one system and some others in the other one. Therefore, only correlative statements can be made, and no conclusive causal relationship can be established. In addition, in several instances, appropriate experimental controls were missing and experiments were poorly executed (please see the specific comments below).

Major concerns

1. Throughout the paper, the authors correlate the high/low mitochondrial membrane potential (MMP) to high/low mitochondrial activity and oxidative phosphorylation (OxPhos). However, high MMP does not always indicate high mitochondrial activity and TCA cycle function. To draw conclusions about mitochondrial function, the authors should perform a full bioenergetics characterisation of the sorted cells, including measuring oxygen consumption rate – OCR, and any latent mitochondrial dysfunction which could result in high MMP, for instance via the reversal of complex V (using oligomycin treatment followed by measurement of MMP).

2. The authors perform tail vein injection after treating cells with specific compounds and conclude that the differences in the metastatic potential of the injected cells are due to the pharmacological/genetic interventions. However, the lower metastasis could also be explained by reduced cell proliferation during the in vitro treatment and/or due to decreased cell survival in the blood circulation. Although the authors gate for live cells, the detail of the sorting strategy is unclear and these experiments should be

repeated, including all necessary controls (for example, measuring cell proliferation and cell death for all treatments, prior to injection), in Figure 2c, 2d, 6e, 6f, 6g etc. Particularly for Figure 4e, where the authors used the BET inhibitor that could lead to cell cycle arrest, the authors need to carefully test potential changes in proliferation and cell cycle.

3. In Figure 1 the authors conclude that MICs rely on OxPhos and TCA cycle activity. However, in Figure 3, inhibition of the TCA cycle using fluoroacetate enhances the metastatic potential of MICs. These are two contradictory results that generate significant discrepancy and confusion.

Likewise, in Figure 2 the authors claim that lipid storage is indispensable for MICs. Nevertheless, in panel 2k, inhibition of lipolysis using ATGL inhibitor (which would increase lipid storage) results to lower metastatic potential. The authors should explain further these contrasting results.

4. Given that this study focuses on acetyl-CoA metabolism, the authors should provide measurements of acetyl-CoA for every treatment they did in Figures 3-5. Similarly, the authors should perform tracing metabolomic experiments using labelled lipids to properly assess changes in lipid uptake, break-down within the TCA cycle and storage. Alternatively, with regards to FAO, the authors should provide cells with BSA-palmitate and measure OCR using the Seahorse.

5. In Figure 3 the authors used fluoroacetate to inhibit TCA cycle, which would lead to citrate accumulation, export to the cytosol and conversion to acetyl-CoA. However, it is not clear how this axis is relevant to MICs, in conditions where the TCA cycle is functional.

6. Given that acetate and etomoxir act independently, affecting enzymes/metabolites that participate in different metabolic pathways, acetate effects should depend on whether cells are treated exclusively with acetate or in combination with etomoxir. Thus, the authors should explain why cells treated with acetate+etomoxir exhibit lower H3K27ac abundance on the promoters of the selected genes in comparison to acetate-treated cells. Indeed, these results seem to suggest that etomoxir somehow impairs the effects of acetate. In addition, the authors should also show the IgV snapshots from cells treated exclusively with etomoxir.

7. In the last figure (6), the authors validate their model using human cancer cells. However, as indicated above, the results are correlative. For instance, they first sort the cells for their MMP, but instead of measuring histone acetylation in these two groups, they assess the metastatic potential in cells sorted for their histone acetylation levels. However, using this approach, it is unclear whether cells with high MMP are those with high H3K27 acetylation. The authors should measure histone acetylation in the cells sorted for their MMP to make sure that the correct causality is established.

Minor concerns

1. Details about the sorting and gating strategy should be provided for all FACS experiments (for example Figure 1a, 1c etc).

2. Number of replicates should be defined in all experiments including RNA-seq.

3. In Figure 1e the authors measure OCR in monolayer and tumorspheres, report higher maximal OCR in tumorspheres and conclude that OCR is required for the metastatic cells. First, it is not clear why authors cultured cells in monolayers and tumorspheres. They should rather repeat this experiment using MIC and non-MIC cells, after sorting them also for their MMP and/or in 4T1 cells with high/low MMP. In addition, opposite to what the authors claim, there is actually no difference between the basal OCR of monolayer- and tumorsphere-cultured cells. The only difference is observed in the maximal OCR and this is likely because the FCCP concentration is too high for the cells cultured in monolayers (as evidenced by the fact that their maximal OCR decreased over the course of the FCCP treatment). Thus, this result might not be reflective of real OCR differences between monolayers and tumorspheres.

4. In Figure 3b and 3c the authors should include the appropriate control to demonstrate that the treatment works as expected (for instance by measuring citrate).

5. In Figure 2a error bars should be added to the graph. Importantly, this graph shows differences in basal OCR between monolayer and tumorsphere in contrast with Fig. 1e.

6. The authors report changes in the lipid metabolism and assess lipid uptake and storage. It would be interesting to investigate how is de novo lipid biosynthesis in MICs and non-MICs.

7. The Oil Red O staining in Figure 2j is of low quality and the experiment needs to be repeated.

8. In Figure 4a, 4b and 6d the authors should normalize the H3K27ac MFI to H3 MFI.

9. In Figure 4h the authors should also show statistics for the etomoxir and etomoxir+acetate comparison.

10. In Figure 6d the y-axis label should change to “MMP MFI”.

Reviewer #2 (Remarks to the Author):

The manuscript describes a very interesting and comprehensive study of the metabolic properties of cells defined as cancer-initiating based on high mitochondrial membrane potential, which is a key indicator mitochondrial activity. Breast cancer cell lines and mouse models were used, cells from the models were sorted by flow cytometry and further used in tail vein injections in mice and in a range of different assays for further characterization of the metastasis initiation cells (MICs) vs non-MICs. The authors find that high metabolic activity is crucial for the metastatic capacity of these MICs and that they depend on oxidative phosphorylation and fatty acid oxidation to generate Acetyl-CoA, which in turn is necessary for histone acetylation. They also find a link to EMT with increased expression of EMT marker genes due to increased histone acetylation in acetate-treated cells. This is a valuable contribution to the growing body of literature on how metastatic cancer cells are able to adapt their metabolism during the metastatic process and thereby enabling them to thrive and survive in new environments.

Specific comments:

The study is carried out using mouse models of breast cancer and breast cancer cell lines; mostly triple negative breast cancer, and the PyMT model is known to generate luminal mammary gland tumors. There is not a clear rationale why breast cancer is used as model, or whether cancer type or cancer subtype is at all important in this context. The metabolic flexibility of cancer cells is often understood as a more general feature of cancer progression, and the manuscript lacks a reflection on the choice of a specific cancer vs. the more universal aspects of cancer metabolism.

The parts on analysis of RNA sequencing data are sparse and need to be described more thoroughly. Was a differential gene expression performed to compare the relevant groups (MICs vs. non-MICs or treated vs-non-treated 4T1 cells)? If so, how? Was a pre-ranked gene set enrichment approach used to identify enriched gene sets within the differentially expressed genes? In the last part of Results where the analysis of human breast cancer is described, the heading states “Human models...etc.”. This needs to be changed to human cell line models or similar, to be a more precise description of the contents of the section.

The analysis of the patient cohorts needs to be improved; which cohorts (in plural) were analyzed? Is it gene expression or protein expression of that is described for CPT1A and ACLY? How are the cut-offs for higher vs lower expression determined? What was the OXPHOS signature that was used for Kaplan-Meier analysis identified?

In general, it is difficult to understand what all the figures show without a proper heading as part of the figure itself. Many figures are showing the same thing, in either 4T1 cells or PyMT cells. Such headings could be incorporated in the figure to ease the interpretation of the figures.

What are the differences between Fig 1c and Suppl. Fig 1b? They look identical.

There are many mistakes in reference to figures throughout and in the legends, examples are: Reference to Fig 2d in line 129; in legend to Fig 2j the description of the images and the quantification is swapped; Fig 2d is not referred to in the text; suppl. Fig. 5 has only one figure and does not need the a); suppl. Fig. 6 has a wrong order of figures and the legends do not fit the figures.

Abbreviations are frequently used and not always written in full when appearing first time. When they are written in full first time, they should be used thereafter (ex. MICs)

The word “metastases” as a plural noun is used frequently when it should be “metastasis” as a concept.

Some of the important information is shown in figures but not described fully in the text. For ex. in the first section, tail vein injections of 4T1 cells generate metastasis but it is not described to which organ. This should be clear from the Results, not just in the figure.

Reviewer #3 (Remarks to the Author):

Young and colleagues explore the metabolic profiles of metastasis-initiating breast cancer cells (MICs) and provide strong evidence that MICs rely on TCA cycle and use fatty acid oxidation as a preferential fuel source. This high mitochondrial activity results in the production of acetyl-CoA, which in turn modulates H3K27ac levels and causes breast cancer cells to obtain a partial EMT phenotype, thus promoting metastasis in mouse models of breast cancer. This is a nicely written, well presented manuscript (apart from lack of detail in the Methods and Figure Legends). The experiments are well conducted with the correct controls generating data of a high standard.

I am not an metabolism expert so I cannot comment on the overall novelty of the findings but I would judge this to be a high quality manuscript with findings that are of broad interest to those in the metastasis and stem cell fields, as well as the metabolism field

Major comments/questions

1: A question that remains to be answered here is whether the high MMP potential is only observed in MICs or is a general feature of every tumor? i.e. do all tumors have a proportion of cells with have high MMP regardless of whether it has the ability to metastasize? This could be readily answered by looking at MCF7 or ZR-75-1 tumors.

2: The authors sort their cells for low, medium and high MMP but they do not show what the distribution of their three different population is in vitro (was it sorted for 3 equal populations?). Please provide a plot showing the distribution of these populations in the cell lines used and in the supplementary material a figure showing the gating strategy for the 4T1 and PyMT cells

3: How dynamic is this high MMP phenotype. For example if the authors re-sorted an MMP-high population 1 week later would it contain solely MMP-high cells or would it revert back to a similar distribution as the original population.

3: In most of their experiments the authors nicely show the effect of their inhibitors on proliferation but not in the case of their JQ1 and C646 inhibitor experiments. Have these experiments been done?

4: There is a great variability in the number of lung macrometastases of the vehicle-treated mice between different groups of the same model – i.e. in Fig. 3c, vehicle-treated mice show an average of 100 lung macrometastases, while vehicle-treated mice in Fig. 3k present an average of ~25 macrometastases. Both groups of FVB mice are injected with PyMT cells tail vein and macrometastases are quantified 5 weeks post-injection. Similarly in the 4T1 experiments in Supplementary Fig. 3b and 3d. This is a big difference - any explanation. Most puzzling is that the authors are able to count >400 macrometastases in some experiments e.g. Supplementary Fig. 3f. The authors stated in the methods that macrometastases were counted using a brightfield microscope if the tumor cells were GFP+. How were they counted otherwise? A more detailed methodology for metastatic quantification is required.

5: In Fig. 5b/c the authors analyze the expression of EMT-related genes. What is puzzling here is that none of the core EMT genes such as Vimentin, Cdh1/2, Twist1/2 and Zeb1/2 etc. are examined. Were these genes also increased (or decreased in the case of Cdh1) or were they unaffected? If the latter, what is the explanation. Do the cells that have been treated with acetate or the sorted high MMP population have more mesenchymal morphologies, increased migration etc.?

6: Supplementary 2c and 2d is confusing. Supplementary Fig. 2d is not referred to in the text. In SFig. 2c the authors say that they resected the tumors just before reaching 1000 mm³ and then examined lung mets 30 days later. But on the SFig 2c it is labelled as tumor weight at sacrifice. Further - they state in the text that mets were reduced despite the tumor weight being comparable - I assume that is because they were all removed at a fixed size. Similarly they say that have similar growth rates - it doesn't look very similar, rather it looks as if the Cpt1a primary tumors grow slower. If the authors want to make such a statement, they should measure the growth rates.

7: The clinical data shown in Fig. 6h is very lackluster and could use some more effort (or delete the panel). (a) the y axis is probability - probability of what - metastatic relapse? survival? Using the automatic best cut off is not good practice - the data should be divided into equal high and low, or equal high, medium and low. But more important this is a mix of all different breast cancer subtypes - is the OxPhos signature an independent prognostic marker or does this just act as a surrogate for poor prognostic breast cancers (e.g. triple negative BC, HER2+ breast cancer, highly proliferative breast cancers).

Minor Comments

Overall the methods and figure legends required more detail. For example

1: For every animal experiment the authors should state whether the cells were GFP tagged and whether they were using WT or DEREK mice (the authors should be applauded for using the DEREK mice).

2: Figure 1e, 2a and S2b are missing a u in the y axes in oxygen consumption rate

3: Line 133 -I assume should be Supplementary Fig. 2d, not Fig. 2d

4: The metastases in Fig. 3j are very difficult to see particularly due to the small size and might be better served in the supplementary material where the image can be enlarged

5: The y-axis legend of Fig. 6d is incorrect. It is comparing H3K27ac high and low populations against H3K27ac MFI according to the axis but in the text it is stated that it is measuring MMP.

6: In the legend of Fig. 4i a concentration of 50 nM of JQ1 is listed while in subpanel j a concentration of 500 nM is listed, is this an error?

7: Supplementary Fig. 6h-j are incorrectly labelled in the text, currently listed as Supplementary Fig. 6d-f.

8: Legend for Fig. 1d doesn't include the fGSEA analysis

9: Supplementary Fig. 6 has more panels than are described in the text or in the figure legends

10: Certain experimental protocols and reagent information are missing from the methods and the figure legends. For example, the protocol related to MIC frequency increase after galactose

supplementation presented in Fig. 1g, the protocols used for the metabolic pathway inhibition as presented in Fig. 3 and the inhibitor incubation time, catalogue number and concentration. The house keeping primer is missing from the list.

REBUTTAL LETTER

Reviewer #1

In the paper entitled “Metabolic dependencies of metastasis-initiating cells in breast cancer” Megan and colleagues investigate the role of mitochondrial function and the ensuing metabolic features of tumour metastasis. In particular, the authors found that cells with high mitochondrial potential exhibit higher metastatic ability and that metastasis depends on fatty acid oxidation (FAO), which in turn, is required for acetyl-CoA production. In addition, they report that acetyl-CoA is used to establish the H3K27ac mark on promoters of EMT-related genes, inducing their expression. Lastly, they conclude that by modulating either the acetyl-CoA-generating pathways or specific H3K27Ac writers/readers they can influence the tumorigenic potential.

Although aimed to address a relevant and timely question in the field, this study suffers from major conceptual and technical limitations that undermine the strength of the conclusions. First, the authors used the two different cell models interchangeably throughout the study and perform some experiments in one system and some others in the other one. Therefore, only correlative statements can be made, and no conclusive causal relationship can be established. In addition, in several instances, appropriate experimental controls were missing and experiments were poorly executed (please see the specific comments below).

Largely all experiments were done in both models, and we now include all of the relevant data for both models in the manuscript. Please find below a table comparing the experiments done in both models:

	PyMT	4T1
Elevated mitochondrial activity in MICs	High MMP in MICs (CD90+) High OCR in MICs High OXPHOS & EMT gene expression in MICs High MICs in galactose vs. glucose containing media Mitochondrial ATP synthesis is not limiting for metastasis	High MMP in MICs (ALDH+) High MMP = High metastatic potential High EMT gene expression in high MMP High metastasis formation in galactose vs. glucose containing media Mitochondrial ATP synthesis is not limiting for metastasis
Elevated lipid flux in MICs	Blocking FAO reduces mets Increased FA uptake and storage in MICs Increased FA dependency in MICs (based on Seahorse) Blocking generation and breakdown of lipid stores inhibits mets	Blocking FAO reduces mets Increased FA uptake and storage in MICs High FA storage = High metastatic potential High lipid droplets in early mets
Pathways that generate acetyl-coA are important for metastasis	Fluoroacetate increases metastasis (and this is blocked by Acly inhibition) Acly inhibition by pharmacological or genetic means reduces metastasis Acetate increases metastasis	Fluoroacetate increases metastasis (and this is blocked by Acly inhibition) Blocking FAO and Acly together blocks metastases and rescues acetate induced increase of metastasis Acetate increases metastasis
Increased H3K27ac in MICs and metastasis	Increased H3K27ac:H3 in MICs Reduced metastasis in blocking H3K27ac reader (via JQ1) Increased H3K27ac in acetate and reduced when blocking FAO and/or Acly	Increased H3K27ac:H3 in MICs Reduced metastasis in blocking H3K27ac reader (via JQ1) Increased H3K27ac in acetate and reduced when blocking FAO and Acly –combining all 3 treatments rescues the phenotype

	MIC frequency is rescued by combining acetate with FAOi or JQ1	Metastasis is rescued by combining acetate with FAOi + Aclyi or JQ1
Metabolic inputs are important for EMT	Acetate increases expression of EMT genes	Acetate increases expression of EMT genes

For the autochthonous PyMT model, some experiments are technically challenging to do since the tumor cells are short-lived ex vivo and the tumors can be quite heterogeneous. Further, as the 4T1 more readily metastasize, it becomes easier to perform experiments where we had limited amounts of cells after FACS sorting for injection into mice.

Major concerns

1. Throughout the paper, the authors correlate the high/low mitochondrial membrane potential (MMP) to high/low mitochondrial activity and oxidative phosphorylation (OxPhos). However, high MMP does not always indicate high mitochondrial activity and TCA cycle function. To draw conclusions about mitochondrial function, the authors should perform a full bioenergetics characterisation of the sorted cells, including measuring oxygen consumption rate – OCR, and any latent mitochondrial dysfunction which could result in high MMP, for instance via the reversal of complex V (using oligomycin treatment followed by measurement of MMP).

We did include OCR measurements using the Seahorse instrument. We further now also show that cells grown as spheres with higher stem-like phenotypes have increased OCR as well as heightened MMP compared to monolayer grown cells with reduced stem-like activity. Moreover, we now provide data which analyze ATP content in cells using a fluorescent ATP sensor. Indeed we find that cells with higher MMP have higher levels of ATP, which confirms that in our system MMP indicates mitochondrial activity.

However, as we have described in the text and in the first supplementary figure, we do not think it is correct to perform Seahorse assays on cells that have just been sorted. We tried and we found that their OCR reduces up to 3-fold compared to prior sorting, making it difficult to know if the results are accurate or just due to an artefact (Supplementary Fig. 1g).

2. The authors perform tail vein injection after treating cells with specific compounds and conclude that the differences in the metastatic potential of the injected cells are due to the pharmacological/genetic interventions. However, the lower metastasis could also be explained by reduced cell proliferation during the in vitro treatment and/or due to decreased cell survival in the blood circulation. Although the authors gate for live cells, the detail of the sorting strategy is unclear and these experiments should be repeated, including all necessary controls (for example, measuring cell proliferation and cell death for all treatments, prior to injection), in Figure 2c, 2d, 6e, 6f, 6g etc. Particularly for Figure 4e, where the authors used the BET inhibitor that could lead to cell cycle arrest, the authors need to carefully test potential changes in proliferation and cell cycle.

We had performed proliferation assays for all treatments, but not all of these were included in the first manuscript. This has now been corrected and the figures have now been updated to include proliferation data for all the treatments used. Overall, we find that after treatment, the cells' proliferative ability does not get hampered by the metabolic inhibitors, epigenetic inhibitors, and knockdown constructs (Supplementary Fig. 2c,d, Supplementary Fig. 3f, Supplementary Fig. 4m-n, Supplementary Fig. 5e).

3. In Figure 1 the authors conclude that MICs rely on OxPhos and TCA cycle activity. However, in Figure 3, inhibition of the TCA cycle using fluoroacetate enhances the metastatic potential of MICs. These are two contradictory results that generate significant discrepancy and confusion.

Likewise, in Figure 2 the authors claim that lipid storage is indispensable for MICs. Nevertheless, in panel 2k, inhibition of lipolysis using ATGL inhibitor (which would increase lipid storage) results to lower metastatic potential. The authors should explain further these contrasting results.

We apologize that this important aspect of our study was not sufficiently well explained. While we show that OXPHOS and TCA cycle are enhanced in MIC cells, we further show that this enhanced

mitochondrial activity (as measured as mitochondrial membrane potential, ATP content, and OCR) is actually used by these cells to fill their cytoplasmic acetyl-CoA stores which relies on the export of citrate from the TCA cycle. Since we measured enhanced OXPHOS activity in MIC cells, this export of citrate does not completely deplete the TCA cycle but apparently only uses a minor fraction of its metabolites to supply citrate to the cytoplasm. Obviously, this citrate export depends on mitochondrial activity, which thereby become measurable proxy of this enhanced citrate usage. Completely blocking the TCA cycle by fluoroacetate makes even more citrate available for export and boosts metastasis formation, but this experiment used short term fluoroacetate treatment and long-term treatment may produce other effects.. Interestingly, it had been shown in a recent publication that blocking Aco2 prevents OXPHOS but does not completely disrupt the TCA cycle, despite it being a canonical TCA cycle enzyme¹. We have now carefully gone through the manuscript to make sure that we always state these dependencies in their full complexity in order to avoid such misunderstandings.

Likewise, lipid storage is enhanced in MICs, but at the same time lipid usage by β -oxidation is enhanced as well. Again, lipid stores are a proxy of this enhanced turnover of lipids in MICs. Simply increasing lipid stores by ATGL inhibitor treatment, which actually prevents usage of lipids from these stores, therefore lowers lipid flux and thereby metastatic activity. We now include that in nutrient depleted conditions, MICs lose lipid stores more rapidly than their non-MIC counterparts (Suppl.Fig, 3c). Similarly, if we additionally block lipid droplet generation, lipid stores are stronger reduced in MICs compared to non-MICs (Suppl.Fig, 3d). On the other hand, if we block lipid droplet breakdown in complete media, there is increased lipid stores but loss of metastatic potential (Fig. 2k, Suppl.Fig, 3e). These data show that there is an increased flux of lipids through the lipid stores in MICs and as mentioned in the results, the metastasis assays show that this flux is important for metastasis. Again, we have gone through the manuscript to better explain these connections and to avoid confusion.

4. Given that this study focuses on acetyl-CoA metabolism, the authors should provide measurements of acetyl-CoA for every treatment they did in Figures 3-5. Similarly, the authors should perform tracing metabolomic experiments using labelled lipids to properly assess changes in lipid uptake, break-down within the TCA cycle and storage. Alternatively, with regards to FAO, the authors should provide cells with BSA-palmitate and measure OCR using the Seahorse.

We have now provided measurements for the acetyl-coA/coA ratio and the acetyl-coA sink acetyl-carnitine for the main treatment of interest (acetate; Suppl.Fig, 4g). Changes in acetyl-coA concentration have already been demonstrated by other groups in response to blocking Cpt1a^{2,3} and Acly⁴. We unfortunately cannot perform tracing experiments on endogenous MICs since sorting severely affects their metabolic health (Suppl.Fig, 1g).

To show that there is an increased flux of lipids through MICs, we now include that in nutrient depleted conditions, MICs lose more lipid stores than their non-MIC counterparts (Suppl.Fig. 3c). Similarly, if we block lipid droplet generation in complete media, there is reduced lipid stores in MICs compared to non-MICs (Suppl.Fig. 3d). On the other hand, if we block lipid droplet breakdown in complete media, there is increased lipid stores in MICs compared to non-MICs (Suppl.Fig. 3e).

These data show that there is an increased flux of lipids through the lipid stores in MICs and as mentioned in the results, the metastasis assays show that this flux is important for metastasis. Consequently, when we block this flux either by preventing lipid storage, or mobilization of lipids from lipid stores, or oxidation of lipids, metastasis formation is reduced.

5. In Figure 3 the authors used fluoroacetate to inhibit TCA cycle, which would lead to citrate accumulation, export to the cytosol and conversion to acetyl-CoA. However, it is not clear how this axis is relevant to MICs, in conditions where the TCA cycle is functional.

This question relates to question 3 where we have already pointed out most of the relevant conclusions. Further, we had shown that inhibition of citrate lyase which converts citrate to acetyl-CoA prevents MIC activity. We further show that the TCA cycle is functional in MICs since OXPHOS is increased, clearly demonstrating that the export of citrate only affects a portion of the total TCA metabolites and does not completely deprive the cycle of citrate. Such experimental inhibitions only reflect the most extreme scenario and show the maximum effects in either stimulating MIC activity (e.g. by fluoroacetate which maximises citrate export) or inhibiting MIC activity (e.g. citrate lyase inhibition which blocks citrate to

acetyl-CoA conversion). The endogenous activity lies in-between with a part of the citrate exported to the cytoplasm.

6. Given that acetate and etomoxir act independently, affecting enzymes/metabolites that participate in different metabolic pathways, acetate effects should depend on whether cells are treated exclusively with acetate or in combination with etomoxir. Thus, the authors should explain why cells treated with acetate+etomoxir exhibit lower H3K27ac abundance on the promoters of the selected genes in comparison to acetate-treated cells. Indeed, these results seem to suggest that etomoxir somehow impairs the effects of acetate. In addition, the authors should also show the IgV snapshots from cells treated exclusively with etomoxir.

In cells, the concentrations of acetyl-CoA are defined by the sum of many activities which generate or consume this metabolite (lipid catabolism, citrate export from mitochondria to the cytoplasm, uptake of short chain fatty acids, recycling of acetyl groups after histone deacetylation, etc.). Here we look at two of these metabolic activities which affect acetyl-CoA, but there are many others and the existence of many compensatory feedback-loops complicates the interpretation of such disturbance studies. As we show, β -oxidation is required for MIC activity and contributes to mitochondrial flux and mitochondrial citrate export to the cytoplasm where it is converted to acetyl-CoA. Acetate can directly be converted to acetyl-CoA, but the capacity for this conversion is not unlimited so that added acetate may not overcome a block of fatty acid oxidation by etomoxir. Therefore the net outcome of such treatments

In the last version of the manuscript, we also only showed 3 gene tracks, that may not be wholly representative of the acetylation profile. To prevent misinterpretations of the data, we now included a EMT gene -wide heatmap to show how H3K27ac peaks respond to the treatments of interest.

As requested, we now include the IgV snapshots from cells treated only by etomoxir. We also added the tracks from cells that have been treated with a combination of etomoxir and SB204990, and the combination of etomoxir plus SB204990 plus acetate.

7. In the last figure (6), the authors validate their model using human cancer cells. However, as indicated above, the results are correlative. For instance, they first sort the cells for their MMP, but instead of measuring histone acetylation in these two groups, they assess the metastatic potential in cells sorted for their histone acetylation levels. However, using this approach, it is unclear whether cells with high MMP are those with high H3K27 acetylation. The authors should measure histone acetylation in the cells sorted for their MMP to make sure that the correct causality is established.

We have now updated this figure to show normalized H3K27 levels to H3 levels in high vs. low MMP cells. We have now also included an analysis of the CCLE dataset looking at H3K27ac levels in metastatic vs. non-metastatic cancer cell lines.

Minor concerns

1. Details about the sorting and gating strategy should be provided for all FACS experiments (for example Figure 1a, 1c etc).

The manuscript has now been updated to include plots of the MMP sorting (Fig. 1a), CD24+/CD90+ MICs in the PyMT model (Suppl.Fig, 1d), the ALDH+ MIC population in the 4T1 model (Suppl.Fig, 1c), and the cell death assay by Annexin V and propidium iodide staining (Suppl.Fig, 1i).

2. Number of replicates should be defined in all experiments including RNA-seq.

The manuscript has now been updated to include this information in the legends.

3. In Figure 1e the authors measure OCR in monolayer and tumorspheres, report higher maximal OCR in tumorspheres and conclude that OCR is required for the metastatic cells. First, it is not clear why authors cultured cells in monolayers and tumorspheres. They should rather repeat this experiment using MIC and non-MIC cells, after sorting them also for their MMP and/or in 4T1 cells with high/low MMP. In addition, opposite to what the authors claim, there is actually no difference between the basal OCR of monolayer- and tumorsphere-cultured cells. The only difference is observed in the maximal OCR and this is likely because the FCCP concentration is too high for the cells cultured in monolayers (as evidenced by the fact

that their maximal OCR decreased over the course of the FCCP treatment). Thus, this result might not be reflective of real OCR differences between monolayers and tumorspheres.

As we mentioned before, we cannot perform metabolic measurements on sorted cells due to their poor condition just afterwards (Suppl.Fig. 1g). Instead, we chose to use tumorspheres as a proxy for MICs since these have been widely accepted as being enriched for highly tumorigenic cells in contrast to monolayer cells. We did fail to explain this in the results section, and have now added this detail.

We did titer the FCCP concentration in the PyMT cells, and the concentration we chose was deemed to be appropriate. We also did not think it was correct to be using two different concentrations of FCCP for either tumorsphere or monolayer cultures in this assay as the results may not be comparable.

4. In Figure 3b and 3c the authors should include the appropriate control to demonstrate that the treatment works as expected (for instance by measuring citrate).

The manuscript has now been updated to include this data (SupplFig.4e).

5. In Figure 2a error bars should be added to the graph. Importantly, this graphs shows differences in basal OCR between monolayer and tumorsphere in contrast with Fig. 1e.

Addition of the error bars, makes the graph difficult to see the dips in OCR after the addition of the inhibitors, as shown below. We therefore decided to leave the left part of Figure 2a unchanged, since the relevant result is shown on the right part including all individual samples.

6. The authors report changes in the lipid metabolism and assess lipid uptake and storage. It would be interesting to investigate how is de novo lipid biosynthesis in MICs and non-MICs.

We have shown increased lipid uptake, flux and oxidation in MICs vs. non-MICs. For us this made lipid de novo synthesis less relevant and this was therefore not included.

7. The Oil Red O staining in Figure 2j is of low quality and the experiment needs to be repeated.

We apologize for the low quality of the images which may have lost intensity during conversion to pdf. We have updated the pdf with higher quality images and now use colors adapted for persons with color vision deficiency in the new version of the manuscript.

8. In Figure 4a, 4b and 6d the authors should normalize the H3K27ac MFI to H3 MFI.

The figures 4a and 4b have now been updated to show this normalization. Fig. 6d has been removed.

9. In Figure 4h the authors should also show statistics for the etomoxir and etomoxir+acetate comparison.

The figure (now Suppl.Fig. 5c) has now been updated to include this statistic.

10. In Figure 6d the y-axis label should change to "MMP MFI".

Thank you for catching this error. Fig. 6d has been removed. We now included an analysis of the CCLE dataset looking at H3K27ac levels in metastatic vs. non-metastatic cancer cell lines which is giving a much broader overlook on metastatic cancer cells than only the 4 examples we had shown previously.

Reviewer #2

The manuscript describes a very interesting and comprehensive study of the metabolic properties of cells defined as cancer-initiating based on high mitochondrial membrane potential, which is a key indicator of mitochondrial activity. Breast cancer cell lines and mouse models were used, cells from the models were sorted by flow cytometry and further used in tail vein injections in mice and in a range of different assays for further characterization of the metastasis initiation cells (MICs) vs non-MICs. The authors find that high metabolic activity is crucial for the metastatic capacity of these MICs and that they depend on oxidative phosphorylation and fatty acid oxidation to generate Acetyl-CoA, which in turn is necessary for histone acetylation. They also find a link to EMT with increased expression of EMT marker genes due to increased histone acetylation in acetate-treated cells. This is a valuable contribution to the growing body of literature on how metastatic cancer cells are able to adapt their metabolism during the metastatic process and thereby enabling them to thrive and survive in new environments.

We thank this reviewer for his excellent summary and the positive evaluation of our work.

Specific comments:

The study is carried out using mouse models of breast cancer and breast cancer cell lines; mostly triple negative breast cancer, and the PyMT model is known to generate luminal mammary gland tumors. There is not a clear rationale why breast cancer is used as model, or whether cancer type or cancer subtype is at all important in this context. The metabolic flexibility of cancer cells is often understood as a more general feature of cancer progression, and the manuscript lacks a reflection on the choice of a specific cancer vs. the more universal aspects of cancer metabolism.

We had chosen to perform this investigation in breast cancer since the lab had identified a highly metastatic cell population in the PyMT model⁵. We used PyMT tumors that are at a late stage, which has been shown to turn into a TNBC model⁶. While the observed influence of metabolism may be applicable to other cancer types, this extension to other types is beyond the scope of the current manuscript.

We have looked at a potential role of H3K27ac in other cancer types using the Cancer Cell Line Encyclopedia and MetMap, and we find a clear positive correlation between this epigenetic mark and metastatic potential only for breast cancer. But we lack further evidence to include this in the manuscript.

The parts on analysis of RNA sequencing data are sparse and need to be described more thoroughly. Was a differential gene expression performed to compare the relevant groups (MICs vs. non-MICs or treated vs-non-treated 4T1 cells)? If so, how? Was a pre-ranked gene set enrichment approach used to identify enriched gene sets within the differentially expressed genes? In the last part of Results where the analysis of human breast cancer is described, the heading states “Human models...etc.”. This needs to

be changed to human cell line models or similar, to be a more precise description of the contents of the section.

As recommended by the GSEA, we did not perform pre-ranking, rather, we have normalized the count matrices, filtered only for genes that are expressed by at least half of the samples, and then loaded the dataset into the GSEA software. We then performed the analysis against the Hallmark and Gene Ontology gene sets. The methods have now been updated to include this detail.

We have now also changed the title of this section as “Human breast cancer cell lines exhibit a similar dependence on mitochondrial activity and acetyl-CoA for metastasis”

The analysis of the patient cohorts needs to be improved; which cohorts (in plural) were analyzed? Is it gene expression or protein expression of that is described for CPT1A and ACLY? How are the cut-offs for higher vs lower expression determined? What was the OXPHOS signature that was used for Kaplan-Meier analysis identified?

We have analyzed the breast cancer microarray cohort available via KM plotter which comprises 50 datasets. We used gene expression for both the OXPHOS signature and the metabolic enzymes of interest, and patients were stratified based on a median or auto cut-offs. How the auto cut-offs are determined are described by a previously published article⁷, but in brief, a Cox regression is computed starting from the highest and lowest quartile. This is done iteratively, and the point at which the p-values are lowest and the hazard ratio is highest is what is chosen as the cut-off point. Nevertheless, we have now updated the plots to only show comparisons between equal sized groups as suggested by one of the other reviewers.

The OXPHOS signature was generated by taking the genes that were identified from the gene set enrichment analysis to be enriched from the HALLMARK_OXIDATIVE_PHOSPHORYLATION and GOBP_OXIDATIVE_PHOSPHORYLATION in our PyMT MICs. The list of microarray probes are now listed in supplementary table 3 and this detail has been added to the methods section.

In general, it is difficult to understand what all the figures show without a proper heading as part of the figure itself. Many figures are showing the same thing, in either 4T1 cells or PyMT cells. Such headings could be incorporated in the figure to ease the interpretation of the figures.

What are the differences between Fig 1c and Suppl. Fig 1b? They look identical.

Thank you for catching that error. We updated the manuscript now to show that mitochondrial membrane potential is heightened in PyMT MICs even after normalizing to mitochondrial mass. We have updated the figures to include this information to make them more easily interpreted. Please note that the two plots look quite similar because the mitochondrial mass between the two populations were also quite similar.

There are many mistakes in reference to figures throughout and in the legends, examples are: Reference to Fig 2d in line 129; in legend to Fig 2j the description of the images and the quantification is swapped; Fig 2d is not referred to in the text; suppl. Fig. 5 has only one figure and does not need the a); suppl. Fig. 6 has a wrong order of figures and the legends do not fit the figures.

Thank you for catching these errors. We have now carefully looked over the manuscript to ensure that the figure references are correct.

Abbreviations are frequently used and not always written in full when appearing first time. When they are written in full first time, they should be used thereafter (ex. MICs)

Thank you for catching this, we have now looked over to ensure that the abbreviations are described first prior to using them.

The word “metastases” as a plural noun is used frequently when it should be “metastasis” as a concept.

Thank you for catching this, we’ve now corrected this error.

Some of the important information is shown in figures but not described fully in the text. For ex. In the first section, tail vein injections of 4T1 cells generate metastasis but it is not described to which organ. This should be clear from the Results, not just in the figure.

This is lung metastasis, We have now updated the results section to include this information.

Reviewer #3

Young and colleagues explore the metabolic profiles of metastasis-initiating breast cancer cells (MICs) and provide strong evidence that MICs rely on TCA cycle and use fatty acid oxidation as a preferential fuel source. This high mitochondrial activity results in the production of acetyl-CoA, which in turn modulates H3K27ac levels and causes breast cancer cells to obtain a partial EMT phenotype, thus promoting metastasis in mouse models of breast cancer. This is a nicely written, well presented manuscript (apart from lack of detail in the Methods and Figure Legends). The experiments are well conducted with the correct controls generating data of a high standard.

I am not an metabolism expert so I cannot comment on the overall novelty of the findings but I would judge this to be a high quality manuscript with findings that are of broad interest to those in the metastasis and stem cell fields, as well as the metabolism field

We thank this reviewer for his excellent summary and the positive evaluation of our work.

Major comments/questions

1: A question that remains to be answered here is whether the high MMP potential is only observed in MICs or is a general feature of every tumor? i.e. do all tumors have a proportion of cells with have high MMP regardless of whether it has the ability to metastasize? This could be readily answered by looking at MCF7 or ZR-75-1 tumors.

Indeed, all cancer cells show a spectrum of individual cellular MMP activity, but we are not sure if we can directly compare MMP levels between cell lines in absolute terms. We have observed that higher mitochondrial activity is a feature of the subpopulation of strongly metastatic cells in the cancer models we have analyzed. Based on additional results we concluded that high mitochondrial activity is a pre-requisite for metastasis, but we cannot conclude if this on its own is sufficient, i.e. whether forcing non-metastatic cell lines to undergo more OXPHOS would render them metastatic.

Interestingly, we find that high expression of genes involved in oxidative phosphorylation predict reduced distant relapse-free survival in breast cancer patients. This measure may be used as equivalent to the cellular MMP measurements and suggests that indeed there may be a direct link between mitochondrial activity and metastatic activity.

2: The authors sort their cells for low, medium and high MMP but they do not show what the distribution of their three different population is in vitro (was it sorted for 3 equal populations?). Please provide a plot showing the distribution of these populations in the cell lines used and in the supplementary material a figure showing the gating strategy for the 4T1 and PyMT cells

We sorted cells based on a cut-off of 5% for each population, i.e. Low: lowest 5%, mid 10-90%, high: above 95%. The gating strategies are now included in the manuscript (Fig. 1a).

3: How dynamic it this high MMP phenotype. For example if the authors re-sorted an MMP-high population 1 week later would it contain solely MMP-high cells or would it revert back to a similar distribution as the original population.

What we have observed is that PyMT MICs maintain their high MMP phenotype for as long as we have them in culture. However, if we sort these MICs and plate them separately, they revert back to the characteristics of the original population.

3: In most of their experiments the authors nicely show the effect of their inhibitors on proliferation but not in the case of their JQ1 and C646 inhibitor experiments. Have these experiments been done?

We have now included this data to the manuscript (Suppl.Fig. 5a, e).

4: There is a great variability in the number of lung macrometastases of the vehicle-treated mice between different groups of the same model – i.e. in Fig. 3c, vehicle-treated mice show an average of 100 lung macrometastases, while vehicle-treated mice in Fig. 3k present an average of ~25 macrometastases. Both groups of FVB mice are injected with PyMT cells tail vein and macrometastases are quantified 5 weeks post-injection. Similarly in the 4T1 experiments in Supplementary Fig. 3b and 3d. This is a big

difference — any explanation. Most puzzling is that the authors are able to count >400 macrometastases in some experiments e.g. Supplementary Fig. 3f. The authors stated in the methods that macrometastases were counted using a brightfield microscope if the the tumor cells were GFP+. How were they counted otherwise? A more detailed methodology for metastatic quantification is required. When we use the PyMT model, we for each experiment obtain fresh tumor cells from spontaneously developing tumors. Even when these are collected from similar aged mice and at similar tumor size, there are differences in the metastatic capacity of these tumor cells between experiments. We therefore always include untreated controls to account for these differences, but this makes direct comparisons in absolute numbers between different experiments impossible.

For the 4T1 cells, we usually get similar number of metastases in the mice if they were injected into syngeneic mice, and while supplementary Fig3d is on the higher side of the number of metastases we can count, it is still within the range of the norm. In our hands, we typically get 25-100 metastases in the control for the 4T1 cells.

We have updated the methods section to better explain the counting procedure.

5: In Fig. 5b/c the authors analyze the expression of EMT-related genes. What is puzzling here is that none of the core EMT genes such as Vimentin, Cdh1/2, Twist1/2 and Zeb1/2 etc. are examined. Were these genes also increased (or decreased in the case of Cdh1) or were they unaffected? If the latter, what is the explanation. Do the cells that have been treated with acetate or the sorted high MMP population have more mesenchymal morphologies, increased migration etc.?

We have observed a tendency to increase Twist2 expression in the acetate treated cells, however, not much change in other EMT genes. This has now been included in the manuscript (Fig. 5b). We tried to measure Twist2 expression in the conditions where JQ1 is used as a treatment, however, the gene becomes undetectable. Cells that have been treated with acetate do show a more mesenchymal phenotype after the treatment period. Finally, cells that are sorted for high MMP show enhanced expression of the gene set for EMT –this has now been included in the manuscript (Suppl.Fig. 6b).

6: Supplementary 2c and 2d is confusing. Supplementary Fig. 2d is not referred to in the text. In SFig. 2c the authors say that they resected the tumors just before reaching 1000 mm³ and then examined lung mets 30 days later. But on the SFig 2c it is labelled as tumor weight at sacrifice. Further - they state in the text that mets were reduced despite the tumor weight being comparable - I assume that is because they were all removed at a fixed size. Similarly they say that have similar growth rates - it doesn't look very similar, rather it looks as if the Cpt1a primary tumors grow slower. If the authors want to make such a statement, they should measure the growth rates.

Thank you for catching these errors. We have now changed the text to better explain the experimental protocol and figure references and we have also corrected the axis titles. Indeed, we did remove the tumors as they reached a certain size, the figure on the right panel of Supplementary Figure 2 was just to demonstrate that the tumor weights did not differ. Since we removed these tumors all at the same size, we waited slightly longer for the Cpt1a KD tumors which gave these tumors more time to seed cells. Nevertheless, the final metastasis numbers for these Cpt1a KD tumors was reduced.

7: The clinical data shown in Fig. 6h is very lackluster and could use some more effort (or delete the panel). (a) the y axis is probability - probability of what - metastatic relapse? survival? Using the automatic best cut off is not good practice - the data should be divided into equal high and low, or equal high, medium and low. But more important this is a mix of all different breast cancer subtypes - is the OxPhos signature an independent prognostic marker or does this just act as a surrogate for poor prognostic breast cancers (e.g. triple negative BC, HER2+ breast cancer, highly proliferative breast cancers).

We have now updated the figure axis and titles to be more informative. We have also now updated the figures so that the comparisons are done between equal sized groups. This cohort is a mix of all breast cancer types, but if we stratify them based on Pam50 classification, the probability of metastatic relapse is increased specifically in basal and her2-positive tumors for high OXPHOS signature expressing tumors.

Minor Comments

Overall the methods and figure legends required more detail. For example

1: For every animal experiment the authors should state whether the cells were GFP tagged and whether they were using WT or DEREK mice (the authors should be applauded for using the DEREK mice. The figure captions have now been updated to include what kind of cells (if, for example, GFP+) into which mouse strain.

2: Figure 1e, 2a and S2b are missing a u in the y axes in oxygen consUmption rate
Thank you for catching the error, the figures are now updated.

3: Line 133 -I assume should be Supplementary Fig. 2d, not Fig. 2d
Thank you for catching the error, we have now updated the text.

4: The metastases in Fig. 3j are very difficult to see particularly due to the small size and might be better served in the supplementary material where the image can be enlarged
Thank you for the suggestion, we have now enlarged the photo (now Fig. 3l).

5: The y-axis legend of Fig. 6d is incorrect. It is comparing H3K27ac high and low populations against H3K27ac MFI according to the axis but in the text it is stated that it is measuring MMP.
Thank you for catching our error. Fig. 6d has been removed. We now included an analysis of the CCLE dataset looking at H3K27ac levels in metastatic vs. non-metastatic cancer cell lines which is giving a much broader overlook on metastatic cancer cells than only the 4 examples we had shown previously.

6: In the legend of Fig. 4i a concentration of 50 nM of JQ1 is listed while in subpanel j a concentration of 500 nM is listed, is this an error?
This is not an error. The PyMT and 4T1 cells have differing sensitivities to the drugs used.

7: Supplementary Fig. 6h-j are incorrectly labelled in the text, currently listed as Supplementary Fig. 6d-f.
Thank you for catching the error, we have now updated the text.

8: Legend for Fig. 1d doesn't include the GSEA analysis
Thank you for catching the error, we have now included a description of the GSEA in the captions.

9: Supplementary Fig. 6 has more panels than are described in the text or in the figure legends
Thank you for catching the error, we have now updated the figure and corresponding captions.

10: Certain experimental protocols and reagent information are missing from the methods and the figure legends. For example, the protocol related to MIC frequency increase after galactose supplementation presented in Fig. 1g, the protocols used for the metabolic pathway inhibition as presented in Fig. 3 and

the inhibitor incubation time, catalogue number and concentration. The house keeping primer is missing from the list.

We have now updated the experimental protocols to include the protocol for galactose supplementation and a table listing all the inhibitors used (with concentration, supplier, and cat number). We have also now added the missing housekeeping gene primers to the list.

REFERENCES:

1. Arnold, P. K. *et al.* A non-canonical tricarboxylic acid cycle underlies cellular identity. *Nature* **603**, 477–481 (2022).
2. Corbet, C. *et al.* Acidosis Drives the Reprogramming of Fatty Acid Metabolism in Cancer Cells through Changes in Mitochondrial and Histone Acetylation. *Cell Metab.* **24**, 311–323 (2016).
3. Altea-Manzano, P. *et al.* A palmitate-rich metastatic niche enables metastasis growth via p65 acetylation resulting in pro-metastatic NF- κ B signaling. *Nat. Cancer* **4**, (2023).
4. Hatzivassiliou, G. *et al.* ATP citrate lyase inhibition can suppress tumor cell growth. *Cancer Cell* **8**, 311–321 (2005).
5. Malanchi, I. *et al.* Interactions between cancer stem cells and their niche govern metastatic colonization. *Nature* **481**, (2012).
6. Lin, E. Y. *et al.* Progression to malignancy in the polyoma middle T oncoprotein mouse breast cancer model provides a reliable model for human diseases. *Am. J. Pathol.* **163**, 2113–26 (2003).
7. Lánczky, A. & Györfy, B. Web-based survival analysis tool tailored for medical research (KMplot): Development and implementation. *J. Med. Internet Res.* **23**, 1–7 (2021).

REVIEWERS' COMMENTS

Reviewer #1 (Remarks to the Author):

The reviewer recognizes the author's significant effort to address the concerns raised in the initial submission. The additional experiments are, in most cases properly controlled and nicely executed, and they generally support the conclusions of the work. However, the reviewer remains unconvinced that the data fully support the proposed mechanistic connection between high MMP, OxPhos activity, ATP production, TCA cycle fluxes, and citrate-specific production. Due to the intrinsic difficulty of performing experiments in sorted MICs, which the referee fully understands, this connection remains mostly correlative and the mechanism is not sufficiently granular. For example, the authors showed that total ATP levels of MICs are higher than non-MICs (Figure S1a); thus, they linked MMP to mitochondria function. However, mitochondria-independent pathways also generate ATP. Therefore, the authors should measure mitochondrial-specific ATP levels, to make the connection between MMP and ATP/mitochondrial function stronger. Furthermore, the authors focused only on the first step (ie., citrate production in mitochondria) and the last step (ie., histone acetylation) of a multi-step process, without investigating changes occurring in all intermediate points, including the specific diversion of carbons from the TCA cycle to citrate, citrate exports, and conversion to AcCoA.

Given (i) the intrinsic limitations of the cell system that the authors use, which makes it difficult to perform some of the previously and currently suggested experiments, and (ii) the amount of data that this manuscript already contains, the reviewer would discourage any further experiments and would thus leave the decision of whether this manuscript should be accepted as is, possibly toning down some of the conclusions regarding the above-described mechanism and whether the depth of investigation is sufficient, to the editor.

Reviewer #2 (Remarks to the Author):

The authors have submitted a revised version of their manuscript "Metabolic dependencies of metastasis-initiating cells in breast cancer" in which they have added new data and edited some of the text according to review comments. They have more thoroughly described details on data analysis, which were lacking in the first version, and they have corrected mistakes that were pointed out by this reviewer. They have also responded specifically to the review comments in a rebuttal document.

The manuscript describes a substantial amount of work and on an important topic.

There are still some issues that need to be resolved:

While they have addressed the reasoning behind using breast cancer as a model in the rebuttal letter, this has not been addressed anywhere in the Introduction. In fact, the title has Breast cancer but no justification or explanation for using breast cancer cell lines in this study is included in the Introduction. In the Discussion, the authors have added “breast cancer “ several times, yet this only specifies that breast cancer cells were used.

The added sentence in line 85-86 lacks a word “compared with” or similar.

Define the abbreviation MFI when first used.

Legend to Figure 2j stills needs correction.

PAM50 should be written with only capital letters.

Reviewer #3 (Remarks to the Author):

I was Reviewer 2 for this manuscript. In this revised version of their manuscript the authors have done a very good job in addressing all of the comments raised at the 1st review including (a) presenting new data and (b) providing all the requested additional technical/method/figure legend information. As a consequence, I would judge this to be a much improved manuscript.

I have no further comments

REVIEWERS' COMMENTS

Reviewer #1 (Remarks to the Author):

The reviewer recognizes the author's significant effort to address the concerns raised in the initial submission. The additional experiments are, in most cases properly controlled and nicely executed, and they generally support the conclusions of the work. However, the reviewer remains unconvinced that the data fully support the proposed mechanistic connection between high MMP, OxPhos activity, ATP production, TCA cycle fluxes, and citrate-specific production. Due to the intrinsic difficulty of performing experiments in sorted MICs, which the referee fully understands, this connection remains mostly correlative and the mechanism is not sufficiently granular. For example, the authors showed that total ATP levels of MICs are higher than non-MICs (Figure S1a); thus, they linked MMP to mitochondria function. However, mitochondria-independent pathways also generate ATP. Therefore, the authors should measure mitochondrial-specific ATP levels, to make the connection between MMP and ATP/mitochondrial function stronger. Furthermore, the authors focused only on the first step (ie., citrate production in mitochondria) and the last step (ie., histone acetylation) of a multi-step process, without investigating changes occurring in all intermediate points, including the specific diversion of carbons from the TCA cycle to citrate, citrate exports, and conversion to AcCoA.

Given (i) the intrinsic limitations of the cell system that the authors use, which makes it difficult to perform some of the previously and currently suggested experiments, and (ii) the amount of data that this manuscript already contains, the reviewer would discourage any further experiments and would thus leave the decision of whether this manuscript should be accepted as is, possibly toning down some of the conclusions regarding the above-described mechanism and whether the depth of investigation is sufficient, to the editor.

We agree to this reviewer that our system of autochthonous tumors and small, subpopulations of cancer cells which need to be isolated by FACS pose limitations on the type of experiments which can be performed. We have now toned down some of our conclusions as suggested by this reviewer.

Reviewer #2 (Remarks to the Author):

The authors have submitted a revised version of their manuscript "Metabolic dependencies of metastasis-initiating cells in breast cancer" in which they have added new data and edited some of the text according to review comments. They have more thoroughly described details on data analysis, which were lacking in the first version, and they have corrected mistakes that were pointed out by this reviewer. They have also responded specifically to the review comments in a rebuttal document.

The manuscript describes a substantial amount of work and on an important topic.

There are still some issues that need to be resolved:

While they have addressed the reasoning behind using breast cancer as a model in the rebuttal letter, this has not been addressed anywhere in the Introduction. In fact, the title has Breast cancer but no justification or explanation for using breast cancer cell lines in this study is included in the Introduction. In the Discussion, the authors have added "breast cancer" several times, yet this only specifies that breast cancer cells were used.

We now added a brief explanation for why we decided to conduct this experiment in breast cancer, similar to what we had written in the initial rebuttal letter.

The added sentence in line 85-86 lacks a word "compared with" or similar.

Thank you for pointing this out, this has now been addressed.

Define the abbreviation MFI when first used.

This is now defined in the figure legends.

Legend to Figure 2j stills needs correction.

This has now been corrected.

PAM50 should be written with only capital letters.

This has now been corrected

Reviewer #3 (Remarks to the Author):

I was Reviewer 2 for this manuscript. In this revised version of their manuscript the authors have done a very good job in addressing all of the comments raised at the 1st review including (a) presenting new data and (b) providing all the requested additional technical/method/figure legend information. As a consequence, I would judge this to be a much improved manuscript.

I have no further comments